# Relative stability toward diffeomorphisms indicates performance in deep nets

**Leonardo Petrini,    Alessandro Favero,    Mario Geiger,    Matthieu Wyart**

Institute of Physics
École Polytechnique Fédérale de Lausanne
1015 Lausanne, Switzerland
{name.surname}@epfl.ch

## Abstract

Understanding why deep nets can classify data in large dimensions remains a challenge. It has been proposed that they do so by becoming stable to diffeomorphisms, yet existing empirical measurements support that it is often not the case. We revisit this question by defining a maximum-entropy distribution on diffeomorphisms, that allows to study typical diffeomorphisms of a given norm. We confirm that stability toward diffeomorphisms does not strongly correlate to performance on benchmark data sets of images. By contrast, we find that the *stability toward diffeomorphisms relative to that of generic transformations* $R_f$ correlates remarkably with the test error $\epsilon_t$. It is of order unity at initialization but decreases by several decades during training for state-of-the-art architectures. For CIFAR10 and 15 known architectures we find $\epsilon_t \approx 0.2\sqrt{R_f}$, suggesting that obtaining a small $R_f$ is important to achieve good performance. We study how $R_f$ depends on the size of the training set and compare it to a simple model of invariant learning.

## 1   Introduction

Deep learning algorithms LeCun et al. (2015) are now remarkably successful at a wide range of tasks Amodei et al. (2016); Huval et al. (2015); Mnih et al. (2013); Shi et al. (2016); Silver et al. (2017). Yet, understanding how they can classify data in large dimensions remains a challenge. In particular, the curse of dimensionality associated with the geometry of space in large dimension prohibits learning in a generic setting Luxburg and Bousquet (2004). If high-dimensional data can be learnt, then they must be highly structured.

A popular idea is that during training, hidden layers of neurons learn a representation Le (2013) that is insensitive to aspects of the data unrelated to the task, effectively reducing the input dimension and making the problem tractable Ansuini et al. (2019); Recanatesi et al. (2019); Shwartz-Ziv and Tishby (2017). Several quantities have been introduced to study this effect empirically. It includes (i) the mutual information between the hidden and visible layers of neurons Saxe et al. (2019); Shwartz-Ziv and Tishby (2017), (ii) the intrinsic dimension of the neural representation of the data Ansuini et al. (2019); Recanatesi et al. (2019) and (iii) the projection of the label of the data on the main features of the network Kopitkov and Indelman (2020); Oymak et al. (2019); Paccolat et al. (2021a), the latter being defined from the top eigenvectors of the Gram matrix of the neural tangent kernel (NTK) Jacot et al. (2018). All these measures support that the neuronal representation of the data indeed becomes well-suited to the task. Yet, they are agnostic to the nature of what varies in the data that need not being represented by hidden neurons, and thus do not specify what it is.

Recently, there has been a considerable effort to understand the benefits of learning features for one-hidden-layer fully connected nets. Learning features can occur and improve performance when the

true function is highly anisotropic, in the sense that it depends only on a linear subspace of the input space Bach (2017); Chizat and Bach (2020); Ghorbani et al. (2019, 2020); Paccolat et al. (2021a); Refinetti et al. (2021); Yehudai and Shamir (2019). For image classification, such an anisotropy would occur for example if pixels on the edge of the image are unrelated to the task. Yet, fully-connected nets (unlike CNNs) acting on images tend to perform best in training regimes where features are not learnt Geiger et al. (2021, 2020); Lee et al. (2020), suggesting that such a linear invariance in the data is not central to the success of deep nets.

Instead, it has been proposed that images can be classified in high dimensions because classes are invariant to smooth deformations or diffeomorphisms of small magnitude Bruna and Mallat (2013); Mallat (2016). Specifically, Mallat and Bruna could handcraft convolution networks, the *scattering transforms*, that perform well and are stable to smooth transformations, in the sense that $\|f(x) - f(\tau x)\|$ is small if the norm of the diffeomorphism $\tau$ is small too. They hypothesized that during training deep nets learn to become stable and thus less sensitive to these deformations, thus improving performance. More recent works generalize this approach to more common CNNs and discuss stability at initialization Bietti and Mairal (2019a,b). Interestingly, enforcing such a stability can improve performance Kayhan and Gemert (2020).

Answering if deep nets become more stable to smooth deformations when trained and quantifying how it affects performance remains a challenge. Recent empirical results revealed that small shifts of images can change the output a lot Azulay and Weiss (2018); Dieleman et al. (2016); Zhang (2019), in apparent contradiction with that hypothesis. Yet in these works, image transformations (i) led to images whose statistics were very different from that of the training set or (ii) were cropping the image, thus are not diffeophormisms. In Ruderman et al. (2018), a class of diffeomorphisms (low-pass filter in spatial frequencies) was introduced to show that stability toward them can improve during training, especially in architectures where pooling layers are absent. Yet, these studies do not address how stability affects performance, and how it depends on the size of the training set. To quantify these properties and to find robust empirical behaviors across architectures, we will argue that the evolution of stability toward smooth deformations needs to be compared relatively to that of any deformation, which turns out to vary significantly during training.

Note that in the context of adversarial robustness, attacks that are geometric transformations of small norm that change the label have been studied Alaifari et al. (2018); Alcorn et al. (2019); Athalye et al. (2018); Engstrom et al. (2019); Fawzi and Frossard (2015); Kanbak et al. (2018); Xiao et al. (2018). These works differ for the literature above and from out study below in the sense that they consider worst-case perturbations instead of typical ones.

## 1.1 Our Contributions

- We introduce a *maximum entropy distribution* of diffeomorphisms, that allow us to generate typical diffeomorphisms of controlled norm. Their amplitude is governed by a "temperature" parameter $T$.

- We define the *relative stability to diffeomorphisms index* $R_f$ that characterizes the square magnitude of the variation of the output function $f$ with respect to the input when it is transformed along a diffeomorphism, relatively to that of a random transformation of the same amplitude. It is averaged on the test set as well as on the ensemble of diffeomorphisms considered.

- We find that at initialization, $R_f$ is close to unity for various data sets and architectures, indicating that initially the output is as sensitive to smooth deformations as it is to random perturbations of the image.

- Our central result is that after training, $R_f$ correlates very strongly with the test error $\epsilon_t$: during training, $R_f$ is reduced by several decades in current State Of The Art (SOTA) architectures on four benchmark datasets including MNIST Lecun et al. (1998), FashionMNIST Xiao et al. (2017), CIFAR-10 Krizhevsky (2009) and ImageNet Deng et al. (2009). For more primitive architectures (whose test error is higher) such as fully connected nets or simple CNNs, $R_f$ remains of order unity. For CIFAR10 we study 15 known architectures and find empirically that $\epsilon_t \approx 0.2\sqrt{R_f}$.

- $R_f$ decreases with the size of the training set $P$. We compare it to an inverse power $1/P$ expected in simple models of invariant learning Paccolat et al. (2021a).

The library implementing diffeomorphisms on images is available online at github.com/pcsl-epfl/diffeomorphism.

The code for training neural nets can be found at github.com/leonardopetrini/diffeo-sota and the corresponding pre-trained models at doi.org/10.5281/zenodo.5589870.

## 2 Maximum-entropy model of diffeomorphisms

### 2.1 Definition of maximum entropy model

We consider the case where the input vector $x$ is an image. It can be thought as a function $x(s)$ describing intensity in position $s = (u, v) \in [0, 1]^2$, where $u$ and $v$ are the horizontal and vertical coordinates. To simplify notations we consider a single channel, in which case $x(s)$ is a scalar (but our analysis holds for colored images as well). We denote by $\tau x$ the image deformed by $\tau$, i.e. $[\tau x](s) = x(s - \tau(s))$. $\tau(s)$ is a vector field of components $(\tau_u(s), \tau_v(s))$. The deformation amplitude is measured by the norm

$$\|\nabla \tau\|^2 = \int_{[0,1]^2} ((\nabla \tau_u)^2 + (\nabla \tau_v)^2) du dv. \tag{1}$$

To test the stability of deep nets toward diffeomorphisms, we seek to build *typical* diffeomorphisms of controlled norm $\|\nabla \tau\|$. We thus consider the distribution over diffeomorphisms that maximizes the entropy with a norm constraint. It can be solved by introducing a Lagrange multiplier $T$ and by decomposing these fields on their Fourier components, see e.g. Kardar (2007) or Appendix A. In this canonical ensemble, one finds that $\tau_u$ and $\tau_v$ are independent with identical statistics. For the picture frame not to be deformed, we impose fixed boundary conditions: $\tau = 0$ if $u = 0, 1$ or $v = 0, 1$. One then obtains:

$$\tau_u = \sum_{i,j \in \mathbb{N}^+} C_{ij} \sin(i\pi u) \sin(j\pi v) \tag{2}$$

where the $C_{ij}$ are Gaussian variables of zero mean and variance $\langle C_{ij}^2 \rangle = T/(i^2 + j^2)$. If the picture is made of $n \times n$ pixels, the result is identical except that the sum runs on $0 < i, j \leq n$. For large $n$, the norm then reads $\|\nabla \tau\|^2 = (\pi^2/2) n^2 T$, and is dominated by high spatial frequency modes. It is useful to add another parameter $c$ to cut-off the effect of high spatial frequencies, which can be simply done by constraining the sum in Eq.2 to $i^2 + j^2 \leq c^2$, one then has $\|\nabla \tau\|^2 = (\pi^3/8) c^2 T$.

Once $\tau$ is generated, pixels are displaced to random positions. A new pixelated image can then be obtained using standard interpolation methods. We use two interpolations, Gaussian and bi-linear[1], as described in Appendix C. As we shall see below, this choice does not affect our result as long as the diffeomorphism induced a displacement of order of the pixel size, or larger. Examples are shown in Fig.1 as a function of $T$ and $c$.

### 2.2 Phase diagram of acceptable diffeomorphisms

Diffeomorphisms are bijective, which is not the case for our transformations if $T$ is too large. When this condition breaks down, a single domain of the picture can break into several pieces, as apparent in Fig.1. It can be expressed as a condition on $\nabla \tau$ that must be satisfied in every point in space Lowe (2004), as recalled in Appendix B. This is satisfied locally with high probability if $\|\tau\|^2 \ll 1$, corresponding to $T \ll (8/\pi^3)/c^2$. In Appendix, we extract empirically a curve of similar form in the $(T, c)$ plane at which a diffeomorphism is obtained with probability at least $1/2$. For much smaller $T$, diffeomorphisms are obtained almost surely.

Finally, for diffeomorphisms to have noticeable consequences, their associated displacement must be of the order of magnitude of the pixel size. Defining $\delta^2$ as the average square norm of the pixel displacement at the center of the image in the unit of pixel size, it is straightforward to obtain from Eq.2 that asymptotically for large $c$ (cf. Appendix B for the derivation),

$$\delta^2 = \frac{\pi}{4} n^2 T \ln(c). \tag{3}$$

The line $\delta = 1/2$ is indicated in Fig.1, using empirical measurements that add pre-asymptotic terms to Eq.3. Overall, the green region corresponds to transformations that (i) are diffeomorphisms with high probability and (ii) produce significant displacements at least of the order of the pixel size.

---

[1] Throughout the paper, if not specified otherwise, bi-linear interpolation is employed.

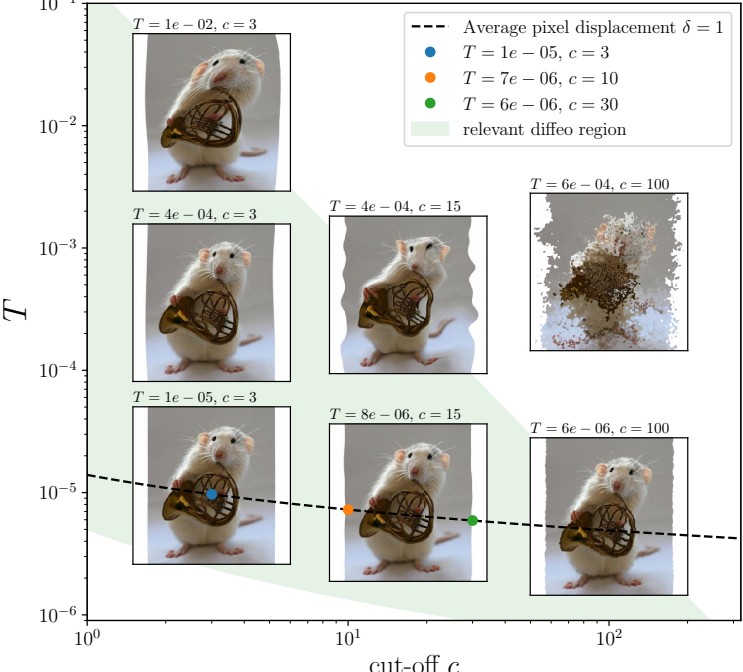

Figure 1: Samples of max-entropy diffeomorphisms for different temperatures $T$ and high-frequency cut-offs $c$ for an ImageNet datapoint of resolution $320 \times 320$. The green region corresponds to well behaving diffeomorphisms (see Section 2.2). The dashed line corresponds to $\delta = 1$. The colored points on the line are those we focus our study in Section 3.

## 3 Measuring the relative stability to diffeomorphisms

**Relative stability to diffeomorphisms** To quantify how a deep net $f$ learns to become less sensitive to diffeomorphisms than to generic data transformations, we define the relative stability to diffeomorphisms $R_f$ as:

$$R_f = \frac{\langle \|f(\tau x) - f(x)\|^2 \rangle_{x,\tau}}{\langle \|f(x + \eta) - f(x)\|^2 \rangle_{x,\eta}}. \tag{4}$$

where the notation $\langle \rangle_y$ can indicate alternatively the mean or the median with respect to the distribution of $y$. In the numerator, this operation is made over the test set and over the ensemble of diffeomorphisms of parameters $(T, c)$ (on which $R_f$ implicitly depends). In the denominator, the average is on the test set and on the vectors $\eta$ sampled uniformly on the sphere of radius $\|\eta\| = \langle \|\tau x - x\| \rangle_{x,\tau}$. An illustration of what $R_f$ captures is shown in Fig.2. In the main text, we consider median quantities, as they reflect better the typical values of distribution. In Appendix E.3 we show that our results for mean quantities, for which our conclusions also apply.

**Dependence of $R_f$ on the diffeomorphism magnitude** Ideally, $R_f$ could be defined for infinitesimal transformations, as it would then characterize the magnitude of the gradient of $f$ along smooth deformations of the images, normalized by the magnitude of the gradient in random directions. However, infinitesimal diffeomorphisms move the image much less than the pixel size, and their definition thus depends significantly on the interpolation method used. It is illustrated in the left panels of Fig.3, showing the dependence of $R_f$ in terms of the diffeomorphism magnitude (here characterised by the mean displacement magnitude at the center of the image $\delta$) for several interpolation methods. We do see that $R_f$ becomes independent of the interpolation when $\delta$ becomes of order unity. In what follows we thus focus on $R_f(\delta = 1)$, which we denote $R_f$.

**SOTA architectures become relatively stable to diffeomorphisms during training, but are not at initialization** The central panels of Fig.3 show $R_f$ at initialization (shaded), and after training (full) for two SOTA architectures on four benchmark data sets. The first key result is that, at initialization, these architectures are as sensitive to diffeomorphisms as they are to random transformations. Relative stability to diffeomorphisms at initialization (guaranteed theoretically in some cases Bietti and Mairal (2019a,b)) thus does not appear to be indicative of successful architectures.

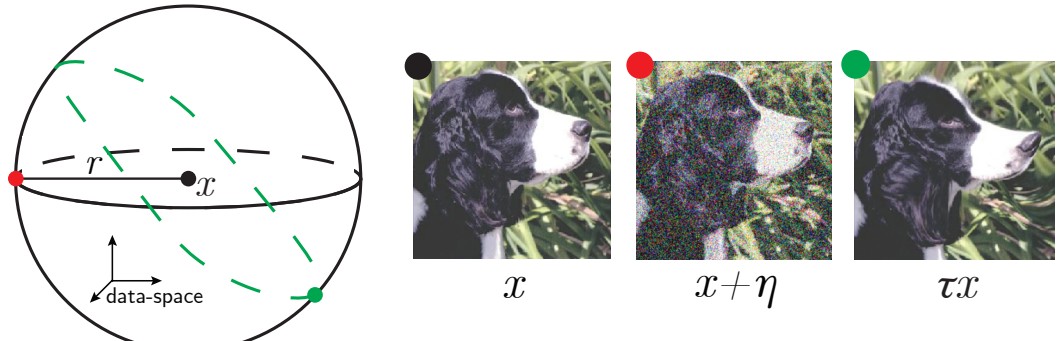

Figure 2: Illustrative drawing of the data-space $\mathbb{R}^{n \times n}$ around a data-point $x$ (black point). We focus here on perturbations of fixed magnitude – i.e. on the sphere of radius $r$ centered in $x$. The intersection between the images of $x$ transformed via typical diffeomorphisms and the sphere is represented in dashed green. By contrast, the red point is an example of random transformation. For large $n$, it is equivalent to adding an i.i.d. Gaussian noise to all the pixel values of $x$. Figures on the right illustrate these transformations, the color of the dot labelling them corresponds to that of the left illustration. The relative stability to diffeomorphisms $R_f$ characterizes how a net $f$ varies in the green directions, normalized by random ones.

By contrast, for these SOTA architectures, relative stability toward diffeomorphisms builds up during training on all the data sets probed. It is a significant effect, with values of $R_f$ after training generally found in the range $R_f \in [10^{-2}, 10^{-1}]$.

Standard data augmentation techniques (translations, crops, and horizontal flips) are employed for training. However, the results we find only mildly depend on using such techniques, see Fig.12 in Appendix.

**Learning relative stability to diffeos requires large training sets**  How many data are needed to learn relative stability toward diffeomorphisms? To answer this question, newly initialized networks are trained on different training-sets of size $P$. $R_f$ is then measured for CIFAR10, as indicated in the right panels of Fig.3. Neural nets need a certain number of training points ($P \sim 10^3$) in order to become relatively stable toward smooth deformations. Past that point, $R_f$ monotonically decreases with $P$. In a range of $P$, this decrease is approximately compatible with the an inverse behavior $R_f \sim 1/P$ found in the simple model of Section 6. Additional results for MNIST and FashionMNIST can be found in Fig.13, Appendix E.3.

**Simple architectures do not become relatively stable to diffeomorphisms**  To test the universality of these results, we focus on two simple architectures: (i) a 4-hidden-layer fully connected (FC) network (FullConn-L4) where each hidden layer has 64 neurons and (ii) LeNet LeCun et al. (1989) that consists of two convolutional layers followed by local max-pooling and three fully-connected layers.

Measurements of $R_f$ for these networks are shown in Fig.4. For the FC net, $R_f \approx 1$ at initialization (as observed for SOTA nets) but *grows* after training on the full data set, showing that FC nets do not learn to become relatively stable to smooth deformations. It is consistent with the modest evolution of $R_f(P)$ with $P$, suggesting that huge training sets would be required to obtain $R_f < 1$. The situation is similar for the primitive CNN LeNet, which only becomes slightly insensitive ($R_f \approx 0.6$) in a single data set (CIFAR10), and otherwise remains larger than unity.

**Layers' relative stability monotonically increases with depth**  Up to this point, we measured the relative stability of the output function for any given architecture. We now study how relative stability builds up as the input data propagate through the hidden layers. In Fig.14 of Appendix E.3, we report $R_f$ as a function of depth for both simple and deep nets. What we observe is $R_{f_0} \approx 1$ independently

---

[2]With the only exception of the ImageNet results (central panel) in which only one trained network is considered.

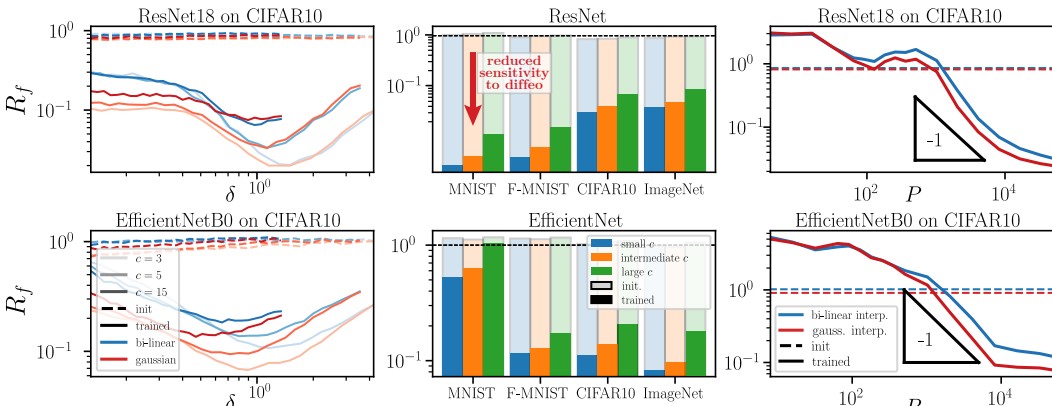

Figure 3: **Relative stability to diffeomorphisms $R_f$ for SOTA architectures.** Left panels: $R_f$ *vs.* diffeomorphism displacement magnitude $\delta$ at initialization (dashed lines) and after training (full lines) on the full data set of CIFAR10 ($P = 50k$) for several cut-off parameters $c$ and two interpolations methods, as indicated in legend. ResNet is shown on the top and EfficientNet on the bottom. Central panels: $R_f(\delta = 1)$ for four different data-sets ($x-$axis) and two different architectures at initialization (shaded histograms) and after training (full histograms). The values of $c$ (in different colors) are $(3, 5, 15)$ and $(3, 10, 30)$ for the first three data-sets and ImageNet, respectively. ResNet18 and EfficientNetB0 are employed for MNIST, F-MNIST and CIFAR10, ResNet101 and EfficientNetB2 for ImageNet. Right panels: $R_f(\delta = 1)$ *vs.* training set size $P$ at $c = 3$ for ResNet18 (top) and EfficientNetB0 (bottom) trained on CIFAR10. The value of $R_{f_0}$ at initialization is indicated with dashed lines. The triangles indicate the predicted slope $R_f \sim P^{-1}$ in a simple model of invariant learning, see Section 6. *Statistics*: Each point in the graphs[2] is obtained by training 16 differently initialized networks on 16 different subsets of the data-sets; each network is then probed with 500 test samples in order to measure stability to diffeomorphisms and Gaussian noise. The resulting $R_f$ is obtained by log-averaging the results from single realizations.

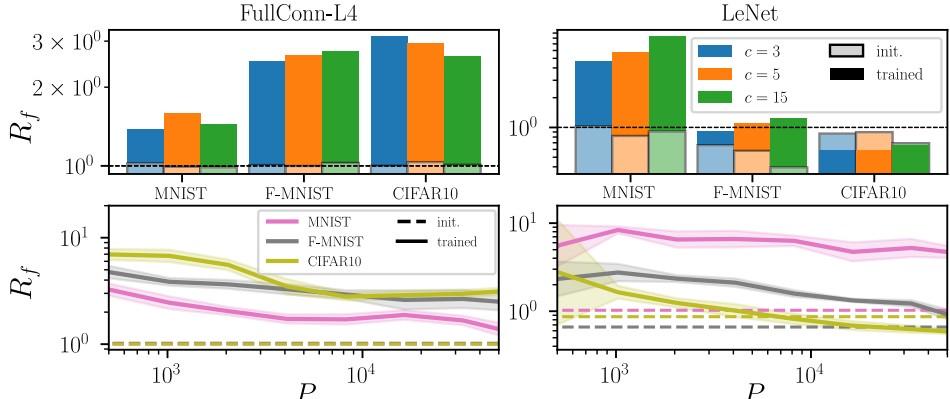

Figure 4: **Relative stability to diffeomorphisms $R_f$ in primitive architectures.** Top panels: $R_f$ at initialization (shaded) or for trained nets (full) for a fully connected net (left) or a primitive CNN (right) at $P = 50k$. Bottom panels: $R_f(P)$ for $c = 3$ and different data sets as indicated in legend. *Statistics:* see caption in the previous figure.

of depth at initialization, and monotonically decreases with depth after training. Overall, the gain in relative stability appears to be well-spread through the net, as is also found for stability alone Ruderman et al. (2018).

# 4 Relative stability to diffeomorphisms indicates performance

Thus, SOTA architectures appear to become relatively stable to diffeomorphisms after training, unlike primitive architectures. This observation suggests that high performance requires such a relative stability to build up. To test further this hypothesis, we select a set of architectures that have been relevant in the state of the art progress over the past decade; we systematically train them in order to compare $R_f$ to their test error $\epsilon_t$. Apart from fully connected nets, we consider the already cited LeNet (5 layers and $\approx 60k$ parameters); then AlexNet Krizhevsky et al. (2012) and VGG Simonyan and Zisserman (2015), deeper (8-19 layers) and highly over-parametrized (10-20M (million) params.) versions of the latter. We introduce *batch-normalization* in VGGs and *skip connections* with ResNets. Finally, we go to EfficientNets, that have all the advancements introduced in previous models and achieve SOTA performance with a relatively small number of parameters (<10M); this is accomplished by designing an efficient small network and properly scaling it up. Further details about these architectures can be found in Table 1, Appendix E.2.

The results are shown in Fig.5. The correlation between $R_f$ and $\epsilon_t$ is remarkably high (corr. coeff.[3] : 0.97), suggesting that generating low relative sensitivity to diffeomorphisms $R_f$ is important to obtain good performance. In Appendix E.3 we also report how changing the train set size $P$ affects the position of a network in the $(\epsilon_t, R_f)$ plane, for the four architectures considered in the previous section (Fig.18). We also show that our results are robust to changes of $\delta$, $c$ (Fig.21) and data sets (Fig.20).

What architectures enable a low $R_f$ value? The latter can be obtained with skip connections or not, and for quite different depths as indicated in Fig.5. Also, the same architecture (EfficientNetB0) trained by transfer learning from ImageNet – instead of directly on CIFAR10 – shows a large improvement both in performance and in diffeomorphisms invariance. Clearly, $R_f$ is much better predicted by $\epsilon_t$ than by the specific features of the architecture indicated in Fig.5.

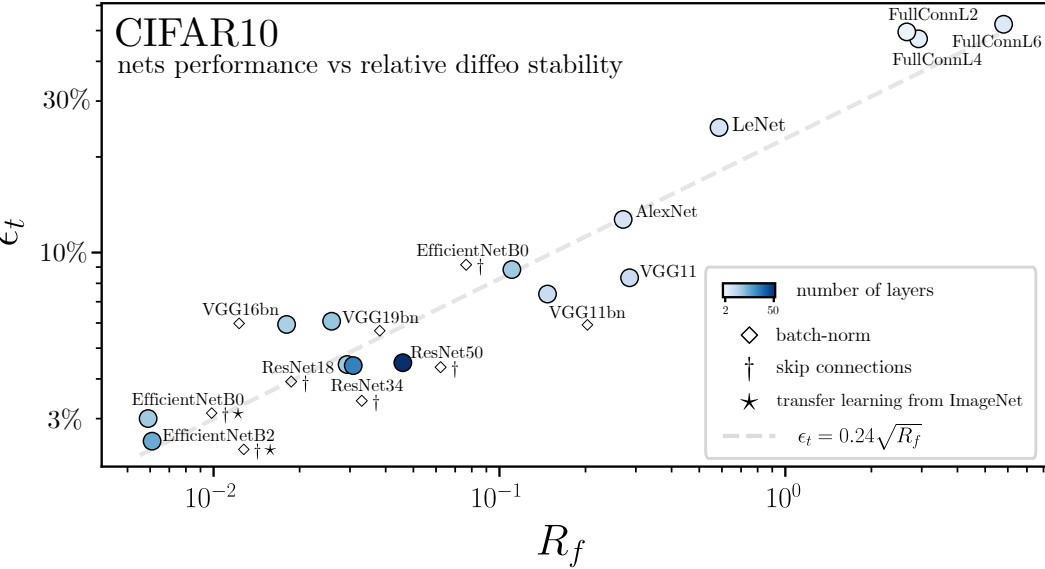

Figure 5: **Test error $\epsilon_t$ *vs.* relative stability to diffeomorphisms** $R_f$ computed at $\delta = 1$ and $c = 3$ for common architectures when trained on the full 10-classes CIFAR10 dataset ($P = 50k$) with SGD and the cross-entropy loss; the EfficientNets achieving the best performance are trained by transfer learning from ImageNet ($\star$) – more details on the training procedures can be found in Appendix E.1. The color scale indicates depth, and the symbols the presence of batch-norm ($\diamond$) and skip connections ($\dagger$). Dashed grey line: power low fit $\epsilon_t \approx 0.2\sqrt{R_f}$. $R_f$ strongly correlates to $\epsilon_t$, much less so to depth or the presence of skip connections. *Statistics:* Each point is obtained by training 5 differently initialized networks; each network is then probed with 500 test samples in order to measure $R_f$. The results are obtained by log-averaging over single realizations. Error bars – omitted here – are shown in Fig.19, Appendix E.3.

# 5 Stability toward diffeomorphisms *vs.* noise

The relative stability to diffeomorphisms $R_f$ can be written as $R_f = {}^{D_f}\!/_{G_f}$ where $G_f$ characterizes the stability with respect to additive noise and $D_f$ the stability toward diffeomorphisms:

$$G_f = \frac{\langle \|f(x+\eta) - f(x)\|^2 \rangle_{x,\eta}}{\langle \|f(x) - f(z)\|^2 \rangle_{x,z}}, \qquad D_f = \frac{\langle \|f(\tau x) - f(x)\|^2 \rangle_{x,\tau}}{\langle \|f(x) - f(z)\|^2 \rangle_{x,z}}. \qquad (5)$$

Here, we chose to normalize these stabilities with the variation of $f$ over the test set (to which both $x$ and $z$ belong), and $\eta$ is a random noise whose magnitude is prescribed as above. Stability toward additive noise has been studied previously in fully connected architectures Novak et al. (2018) and for CNNs as a function of spatial frequency in Tsuzuku and Sato (2019); Yin et al. (2019).

The decrease of $R_f$ with growing training set size $P$ could thus be due to an increase in the stability toward diffeomorphisms (i.e. $D_f$ decreasing with $P$) or a decrease of stability toward noise ($G_f$ increasing with $P$). To test these possibilities, we show in Fig.6 $G_f(P), D_f(P)$ and $R_f(P)$ for MNIST, Fashion MNIST and CIFAR10 for two SOTA architectures. The central results are that (i) stability toward noise is always reduced for larger training sets. This observation is natural: when more data needs to be fitted, the function becomes rougher. (ii) Stability toward diffeomorphisms does not behave universally: it can increase with $P$ or decrease depending on the architecture and the training set. Additionally, $G_f$ and $D_f$ alone show a much smaller correlation with performance than $R_f$– see Figs.15,16,17 in Appendix E.3.

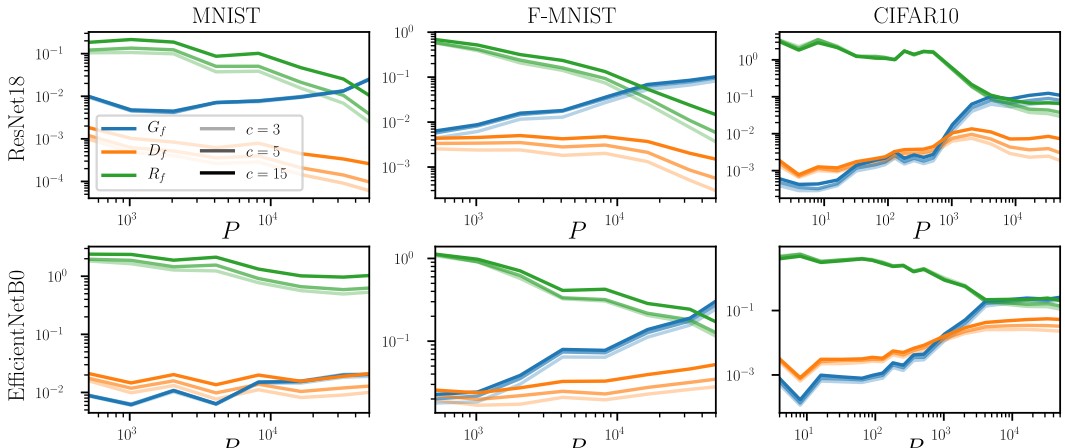

Figure 6: **Stability toward Gaussian noise ($G_f$) and diffeomorphisms ($D_f$) alone, and the relative stability $R_f$.** Columns correspond to different data-sets (MNIST, FashionMNIST and CIFAR10) and rows to architectures (ResNet18 and EfficientNetB0). Each panel reports $G_f$ (blue), $D_f$ (orange) and $R_f$ (green) as a function of $P$ and for different cut-off values $c$, as indicated in the legend. *Statistics:* cf. caption in Fig.3. Error bars – omitted here – are shown in Fig.22, Appendix E.3.

# 6 A minimal model for learning invariants

In this section, we discuss the simplest model of invariance in data where stability to transformation builds up, that can be compared with our observations of $R_f$ above. Specifically, we consider the "stripe" model Paccolat et al. (2021b), corresponding to a binary classification task for Gaussian-distributed data points $x = (x_\parallel, x_\perp)$ where the label function depends only on one direction in data space, namely $y(x) = y(x_\parallel)$. Layers of $y = +1$ and $y = -1$ regions alternate along the direction $x_\parallel$, separated by parallel planes. Hence, the data present $d-1$ invariant directions in input-space denoted by $x_\perp$ as illustrated in Fig.7-left.

When this model is learnt by a one-hidden-layer fully connected net, the first layer of weights can be shown to align with the informative direction Paccolat et al. (2021a). The projection of these weights

---

[3]Correlation coefficient: $\frac{\text{Cov}(\log \epsilon_t, \log R_f)}{\sqrt{\text{Var}(\log \epsilon_t)\text{Var}(\log R_f)}}$.

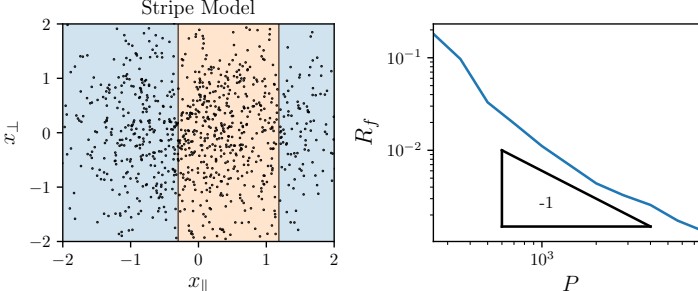

Figure 7: Left: example of the stripe model. Dots are data-points, the vertical lines represent the decision boundary and the color the class label. Right: Relative stability $R_f$ for the stripe model in $d = 30$. The slope of the curve is $-1$, as predicted.

on the orthogonal space vanishes with the training set size $P$ as $1/\sqrt{P}$, an effect induced by the sampling noise associated to finite training sets.

In this model, $R_f$ can be defined as:

$$R_f = \frac{\langle \|f(x_\parallel, x_\perp + \nu) - f(x_\parallel, x_\perp)\|^2 \rangle_{x,\nu}}{\langle \|f(x + \eta) - f(x)\|^2 \rangle_{x,\eta}}, \tag{6}$$

where we made explicit the dependence of $f$ on the two linear subspaces. Here, the isotropic noise $\nu$ is added only in the invariant directions. Again, we impose $\|\eta\| = \|\nu\|$. $R_f(P)$ is shown in Fig. 7-right. We observe that $R_f(P) \sim P^{-1}$, as expected from the weight alignment mentioned above.

Interestingly, Fig.3 for CIFAR10 and SOTA architectures support that the $1/P$ behavior is compatible with the observations for some range of $P$. In Appendix E.3, Fig.13, we show analogous results for MNIST and Fashion-MNIST. We observe the $1/P$ power-law scaling for ResNets. It suggests that for these architectures, learning to become invariant to diffeomorphisms may be limited by a naive measure of sampling noise as well. By contrast for EfficientNets, in which the decrease in $R_f$ is more limited, a $1/P$ behavior cannot be identified.

## 7 Discussion

A common belief is that stability to random noise (small $G_f$) and to diffeomorphisms (small $D_f$) are desirable properties of neural nets. Its underlying assumption is that the true data label mildly depends on such transformations when they are small. Our observations suggest an alternative view:

1. Figs.6,16: better predictors are more sensitive to small perturbations in input space.

2. As a consequence, the notion that predictors are especially insensitive to diffeomorphisms is not captured by stability alone, but rather by the relative stability $R_f = {}^{D_f}/_{G_f}$.

3. We propose the following interpretation of Fig.5: to perform well, the predictor must build large gradients in input space near the decision boundary – leading to a large $G_f$ overall. Networks that are relatively insensitive to diffeomorphisms (small $R_f$) can discover with less data that strong gradients must be there and generalize them to larger regions of input space, improving performance and increasing $G_f$.

This last point can be illustrated in the simple model of Section 6, see Fig.7-left panel. Imagine two data points of different labels falling close to the – e.g. – left true decision boundary. These two points can be far from each other if their orthogonal coordinates differ. Yet, if $R_f = 0$ (now defined in Eq.6), then the output does not depend on the orthogonal coordinates, and it will need to build a strong gradient – in input space – along the parallel coordinate to fit these two data. This strong gradient will exist throughout that entire decision boundary, improving performance but also increasing $G_f$. Instead, if $R_f = 1$, fitting these two data will not lead to a strong gradient, since they can be far from each other in input space. Beyond this intuition, in this model decreasing $R_f$ can quantitatively be shown to increase performance, see Paccolat et al. (2021b).

## 8    Conclusion

We have introduced a novel empirical framework to characterize how deep nets become invariant to diffeomorphisms. It is jointly based on a maximum-entropy distribution for diffeomorphisms, and on the realization that stability of these transformations relative to generic ones $R_f$ strongly correlates to performance, instead of just the diffeomorphisms stability considered in the past.

The ensemble of smooth deformations we introduced may have interesting applications. It could serve as a complement to traditional data-augmentation techniques (whose effect on relative stability is discussed in Fig.12 of the Appendix). A similar idea is present in Hauberg et al. (2016); Shen et al. (2020) but our deformations have the advantage of being easier to sample and data agnostic. Moreover, the ensemble could be used to build adversarial attacks along smooth transformations, in the spirit of Alaifari et al. (2018); Engstrom et al. (2019); Kanbak et al. (2018). It would be interesting to test if networks robust to such attacks are more stable in relative terms, and how such robustness affects their performance.

Finally, the tight correlation between relative stability $R_f$ and test error $\epsilon_t$ suggests that if a predictor displays a given $R_f$, its performance may be bounded from below. The relationships we observe $\epsilon_t(R_f)$ may then be indicative of this bound, which would be a fundamental property of a given data set. Can it be predicted in terms of simpler properties of the data? Introducing simplified models of data with controlled stability to diffeomorphisms beyond the toy model of Section 6 would be useful to investigate this key question.

## Acknowledgements

We thank Alberto Bietti, Joan Bruna, Francesco Cagnetta, Pascal Frossard, Jonas Paccolat, Antonio Sclocchi and Umberto M. Tomasini for helpful discussions. This work was supported by a grant from the Simons Foundation (#454953 Matthieu Wyart).

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
