## A    Maximum entropy calculation

Under the constraint on the borders, $\tau_u$ and $\tau_v$ can be expressed in a real Fourier basis as in Eq.2. By injecting this form into $\|\nabla\tau\|^2$ we obtain:

$$\|\nabla\tau\|^2 = \frac{\pi^2}{4}\sum_{i,j\in\mathbb{N}^+}(C_{ij}^2 + D_{ij}^2)(i^2 + j^2) \tag{7}$$

where $D_{ij}$ are the Fourier coefficients of $\tau_v$. We aim at computing the probability distributions that maximize their entropy while keeping the expectation value of $\|\nabla\tau\|^2$ fixed. Since we have a sum of quadratic random variables, the equipartition theorem Beale (1996) applies: the distributions are normal and every quadratic term contributes in average equally to $\|\nabla\tau\|^2$. Thus, the variance of the coefficients follows $\frac{T}{i^2+j^2}$ where the parameter $T$ determines the magnitude of the diffeomorphism.

## B    Boundaries of studied diffeomorphisms

**Average pixel displacement magnitude $\delta$**    We derive here the large-$c$ asymptotic behavior of $\delta$ (Eq.3). This is defined as the average square norm of the displacement field, in pixel units:

$$\delta^2 = n^2 \int_{[0,1]^2} \|\tau(u,v)\|^2 dudv$$

$$= 2Tn^2 \sum_{i^2+j^2\leq c^2} \frac{1}{i^2+j^2}\int_{[0,1]^2}\sin^2(i\pi u)\sin^2(j\pi v)dudv$$

$$= \frac{Tn^2}{2}\sum_{i^2+j^2\leq c^2}\frac{1}{i^2+j^2}$$

$$\approx \frac{Tn^2}{2}\int_{1\leq x^2+y^2\leq c^2}\frac{1}{x^2+y^2}dxdy$$

$$= \frac{\pi Tn^2}{4}\int_1^c \frac{1}{r}dr$$

$$= \frac{\pi}{4}n^2 T\log c,$$

where we approximated the sum with an integral, in the third step. The asymptotic relations for $\|\nabla\tau\|$ that are reported in the main text are computed in a similar fashion. In Fig.8, we check the agreement between asymptotic prediction and empirical measurements. If $\delta \ll 1$, our results strongly depend on the choice of interpolation method. To avoid it, we only consider conditions for which $\delta \geq 1/2$, leading to

$$T > \frac{1}{\pi n^2 \log c}. \tag{8}$$

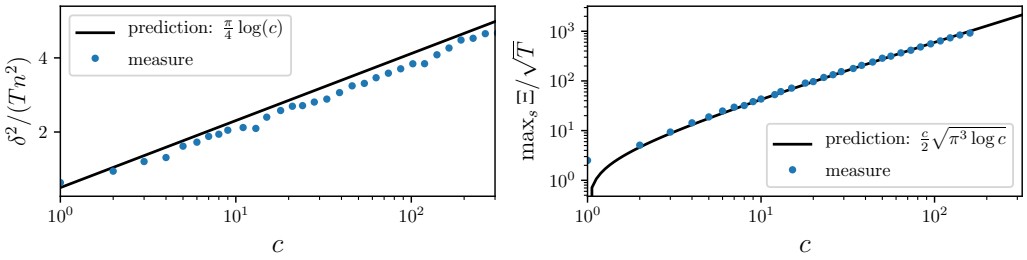

Figure 8: Left: The characteristic displacement $\delta(c,T)$ is observed to follow $\delta^2 \simeq \frac{\pi}{4}n^2 T\log c$. Right: measurement of $\max_s \Xi$ supporting Eq.13.

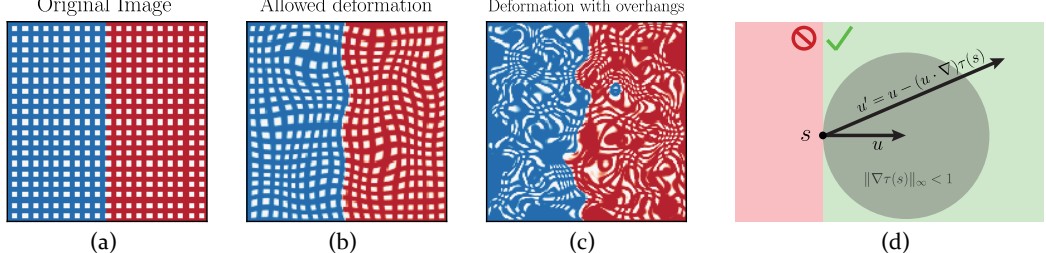

Figure 9: (a) Idealized image at $T = 0$. (b) Diffeomorphism of the image. (c) Deformation of the image at large $T$: colors get mixed-up together, shapes are not preserved anymore. (d) Allowed region for vector transformations under $\tau$. For any point in the image $s$ and any direction $u$, only displacement fields for which all the deformed direction $u'$ is non-zero generate diffeomorphisms. The bound in Eq.12 ($u' \cdot u > 0$) correspond to the green region. The gray disc corresponds to the bound $\|\nabla\tau\|_\infty < 1$.

**Condition for diffeomorphism in the $(T, c)$ plane** For a given value of $c$, there exists a temperature scale beyond which the transformation is not injective anymore, affecting the topology of the image and creating spurious boundaries, see Fig.9a-c for an illustration. Specifically, consider a curve passing by the point $s$ in the deformed image. Its tangent direction is $u$ at the point $s$. When going back to the original image ($s' = s - \tau(s)$) the curve gets deformed and its tangent becomes

$$u' = u - (u \cdot \nabla)\tau(s). \tag{9}$$

A smooth deformation is bijective iff all deformed curves remain curves which is equivalent to have non-zero tangents everywhere

$$\forall\, s, u \neq 0 \quad \|u'\| \neq 0. \tag{10}$$

Imposing $\|u'\| \neq 0$ does not give us any constraint on $\tau$. Therefore, we constraint $\tau$ a bit more and allow only displacement fields such that $u \cdot u' > 0$, which is a sufficient condition for Eq.10 to be satisfied – cf. Fig. 9d. By extremizing over $u$, this condition translates into

$$\frac{1}{2}\left( \sqrt{(\partial_x\tau_x - \partial_y\tau_y)^2 + (\partial_x\tau_y + \partial_y\tau_x)^2} - \partial_x\tau_x - \partial_y\tau_y \right) < 1 \tag{11}$$

or, equivalently,

$$\Xi = \frac{1}{2}\left( \sqrt{\|\nabla\tau\|^2 - 2\det(\nabla\tau)} - \mathrm{Tr}(\nabla\tau) \right) < 1, \tag{12}$$

were we identified by $\Xi$ the l.h.s. of the inequality. We find that the median of the maximum of $\Xi$ over all the image ($\|\Xi(s)\|_\infty$) can be approximated by (see Fig.8b):

$$\max_s \Xi \simeq \frac{c}{2}\sqrt{\pi^3 T \log c}. \tag{13}$$

The resulting constraint on $T$ reads

$$T < \frac{4}{\pi^3 c^2 \log c}. \tag{14}$$

## C  Interpolation methods

When a deformation is applied to an image $x$, each of its pixels gets mapped, from the original pixels grid, to new positions generally outside of the grid itself – cf. Fig. 9a-b. A procedure (interpolation method) needs to be defined to project the deformed image back into the original grid.

For simplicity of notation, we describe interpolation methods considering the square $[0, 1]^2$ as the region in between four pixels – see an illustration in Fig. 10a. We propose here two different ways to interpolate between pixels and then check that our measurements do not depend on the specific method considered.

**Bi-linear Interpolation** The bi-linear interpolation consists, as the name suggests, of two steps of linear interpolation, one on the horizontal, and one on the vertical direction – Fig. 10b. If we look at the square $[0, 1]^2$ and we apply a deformation $\tau$ such that $(0, 0) \mapsto (u, v)$, we have

$$x(u, v) = x(0, 0)(1 - u)(1 - v) + x(1, 0)u(1 - v) + x(0, 1)(1 - u)v + x(1, 1)uv. \tag{15}$$

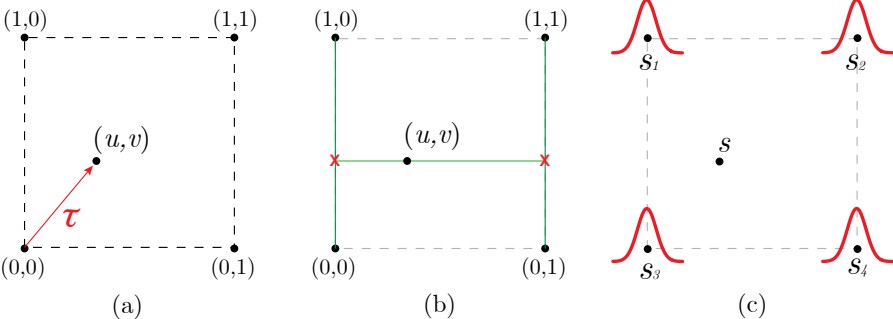

(a)            (b)            (c)

Figure 10: (a) We consider the region between four pixels as the square $[0,1]^2$ where, after the application of a deformation $\tau$, the pixel $(0,0)$ is mapped into $(u,v)$. (b) **Bi-linear interpolation**: the value of $x$ in $(u,v)$ is computed by two steps of linear interpolation. First, we compute $x$ in the red crosses, by averaging values on the vertical axis. Then, a line interpolates horizontally the values in the red crosses to give the result. (c) **Gaussian interpolation**: we denote by $s_i$ the pixel positions in the original grid. The interpolated value of $s$ in any point of the image is given by a weighted sum of $n \times n$ Gaussian centered in each $s_i$ – in red.

**Gaussian Interpolation**    In this case, a Gaussian function[4] is placed on top of each point in the grid – cf. Fig.10. The pixel intensity $x$ can be evaluated at any point outside the grid by computing

$$x(s) = \frac{\sum_i x(s_i) G(s - s_i)}{\sum_i G(s - s_i)}. \tag{16}$$

In order to fix the standard deviation $\sigma$ of $G$, we introduce the *participation ratio* $n$. Given $\Psi_i = G(s, s_i)|_{s=(0.5, 0.5)}$, we define

$$n = \frac{\left( \sum_i \Psi_i^2 \right)^2}{\sum_i \Psi_i^4}. \tag{17}$$

The participation ratio is a measure of how many pixels contribute to the value of a new pixel, which results from interpolation. We fix $\sigma$ in such a way that the participation ratio for the Gaussian interpolation matches the one for the bi-linear ($n = 4$), when the new pixel is equidistant from the four pixels around. This gives $\sigma = 0.4715$.

Notice that this interpolation method is such that it applies a Gaussian smoothing of the image even if $\tau$ is the identity. Consequently, when computing observables for $f$ with the Gaussian interpolation, we always compare $f(\tau x)$ to $f(\tilde{x})$, where $\tilde{x}$ is the smoothed version of $x$, in such a way that $f(\tau^{[T=0]} x) = f(\tilde{x})$.

**Empirical results dependence on interpolation**    Finally, we checked to which extent our results are affected by the specific choice of interpolation method. In particular, blue and red colors in Figs3, 13 correspond to bi-linear and Gaussian interpolation, respectively. The interpolation method only affects the results in the small displacement limit ($\delta \to 0$).

Note: throughout the paper, if not specified otherwise, bi-linear interpolation is employed.

---

[4]$G(s) = (2\pi\sigma^2)^{-1/2} e^{-s^2/2\sigma^2}$.

## D  Stability to additive noise *vs.* noise magnitude

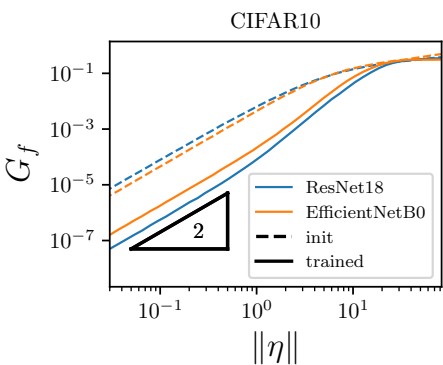 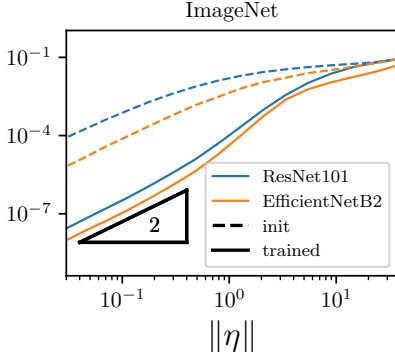

Figure 11: **Stability to isotropic noise** $G_f$ as a function of the noise magnitude $\|\eta\|$ for CIFAR10 (left) and ImageNet (right). The color corresponds to two different classes of SOTA architecture: ResNet and EfficientNet. The slope 2 at small $\|\eta\|$ identifies the linear regime. For larger noise magnitudes, non-linearities appear.

We introduced in Section 5 the stability toward additive noise:

$$G_f = \frac{\langle\|f(x+\eta) - f(x)\|^2\rangle_{x,\eta}}{\langle\|f(x) - f(z)\|^2\rangle_{x,z}}. \tag{18}$$

We study here the dependence of $G_f$ on the noise magnitude $\|\eta\|$. In the $\eta \to 0$ limit, we expect the network function to behave as its first-order Taylor expansion, leading to $G_f \propto \|\eta\|^2$. Hence, for small noise, $G_f$ gives an estimate of the average magnitude of the gradient of $f$ in a random direction $\eta$.

**Empirical results**  Measurements of $G_f$ on SOTA nets trained on benchmark data-sets are shown in Figure 11. We observe that the effect of non-linearities start to be significant around $\|\eta\| = 1$. For large values of the noise – i.e. far away from data-points – the average gradient of $f$ does not change with training.

## E  Numerical experiments

In this Appendix, we provide details on the training procedure, on the different architectures employed and some additional experimental results.

### E.1  Image classification training set-up:

- ○ Trainings are performed in `PyTorch`, the code can be found here github.com/leonardopetrini/diffeo-sota.
- ○ Loss function: cross-entropy.
- ○ Batch size: 128.
- ○ Dynamics:
  - – Fully connected nets: ADAM with `learning rate` $= 0.1$ and no scheduling.
  - – Transfer learning: SGD with `learning rate` $= 10^{-2}$ for the last layer and $10^{-3}$ for the rest of the network, `momentum` $= 0.9$ and `weight decay` $= 10^{-3}$. Both learning rates decay exponentially during training with a factor $\gamma = 0.975$.
  - – All the other networks are trained with SGD with `learning rate` $= 0.1$, `momentum` $= 0.9$ and `weight decay` $= 5 \times 10^{-4}$. The learning rate follows a cosine annealing scheduling Loshchilov and Hutter (2016).

- Early-stopping is performed – i.e. results shown are computed with the network obtaining the best validation accuracy out of 250 training epochs.
- For the experiments involving a training on a subset of the training date of size $P < P_{\max}$, the total number of epochs is accordingly re-scaled in order to keep constant the total number of optimizer steps.
- Standard data augmentation is employed: different random translations and horizontal flips of the input images are generated at each epoch. As a safety check, we verify that the invariance learnt by the nets is not purely due to such augmentation (Fig.12).
- Experiments are run on 16 GPUs NVIDIA V100. Individual trainings run in $\sim 1$ hour of wall time. We estimate a total of a few thousands hours of computing time for running the preliminary and actual experiments present in this work.

The stripe model is trained with an approximation of gradient flow introduced in Geiger et al. (2020), see Paccolat et al. (2021a) for details.

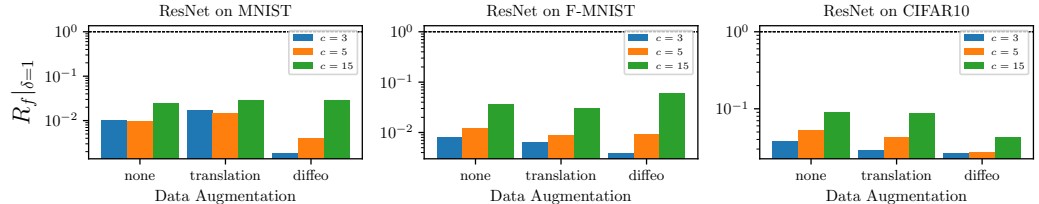

Figure 12: **Effect of data augmentation on $R_f$.** Relative stability to diffeomorphisms $R_f$ after training with different data augmentations: "none" (1st group of bars in each plot) for no data augmentation, "translation" (2nd bars) corresponds to training on randomly translated (by 4 pixels) and cropped inputs, and "diffeo" (3rd bars) to training on randomly deformed images with max-entropy diffeomorphisms ($T = 10^{-2}$, $c = 1$). Results are averaged over 5 trainings of ResNet18 on MNIST (left), FashionMNIST (center), CIFAR10 (right). Colors indicate different cut-off values when probing the trained networks. Different augmentations have a small quantitative, and no qualitative effect on the results. As expected, augmenting the input images with smooth deformations makes the net more invariant to such transformations.

**A note on computing stabilities at init. in presence of batch-norm**  We recall that batch-norm (BN) can work in either of two modes: *training* and *evaluation*. During training, BN computes the mean and variance on the current batch and uses them to normalize the output of a given layer. At the same time, it keeps memory of the running statistics on such batches, and this is used for the normalization steps at inference time (evaluation mode). When probing a network at initialization for computing stabilities, we put the network in evaluation mode, except for batch-norm (BN), which operates in train mode. This is because BN running mean and variance are initialized to 0 and 1, in such a way that its evaluation mode at initialization would correspond to not having BN at all, compromising the input signal propagation in deep architectures.

## E.2 Networks architectures

All networks implementations can be found at github.com/leonardopetrini/diffeo-sota/tree/main/models. In Table 1, we report salient features of the network architectures considered.

Table 1: **Network architectures, main characteristics.** We list here (columns) the classes of net architectures used throughout the paper specifying some salient features (depth, number of parameters, etc...) for each of them.

| features | FullConn | LeNet
LeCun et al. (1989) | AlexNet
Krizhevsky et al. (2012) |
|---|---|---|---|
| depth | 2, 4, 6 | 5 | 8 |
| num. parameters | 200k | 62k | 23 M |
| FC layers | 2, 4, 6 | 3 | 3 |
| activation | ReLU | ReLU | ReLU |
| pooling | / | max | max |
| dropout | / | / | yes |
| batch norm | / | / | / |
| skip connections | / | / | / |

| features | VGG
Simonyan and Zisserman (2015) | ResNet
He et al. (2016) | EfficientNetB0-2
Tan and Le (2019) |
|---|---|---|---|
| depth | 11, 16, 19 | 18, 34, 50 | 18, 25 |
| num. parameters | 9-20 M | 11-24 M | 5, 9 M |
| FC layers | 1 | 1 | 1 |
| activation | ReLU | ReLU | swish |
| pooling | max | avg. (last layer only) | avg. (last layer only) |
| dropout | / | / | yes + dropconnect |
| batch norm | if 'bn' in name | yes | yes |
| skip connections | / | yes | yes (inv. residuals) |

### E.3 Additional figures

We present here:

- ○ Fig.13: $R_f$ as a function of $P$ for MNIST and FashionMNIST with the corresponding predicted slope, omitted in the main text.
- ○ Fig.14: Relative diffeomorphisms stability $R_f$ as a function of depth for simple and deep nets.
- ○ Figs15,16: diffeomorphisms and inverse of the Gaussian stability $D_f$ and $1/G_f$ *vs.* test error for CIFAR10 and the set of architectures considered in Section 4.
- ○ Fig.17: $D_f$, $1/G_f$ and $R_f$ when using the mean in place of the median for computing averages $\langle\cdot\rangle$.
- ○ Fig.18: curves in the $(\epsilon_t, R_f)$ plane when varying the training set size $P$ for FullyConnL4, LeNet, ResNet18 and EfficientNetB0.
- ○ Figs19, 22: error estimates for the main quantities of interest – often omitted in the main text for the sake of figures' clarity.

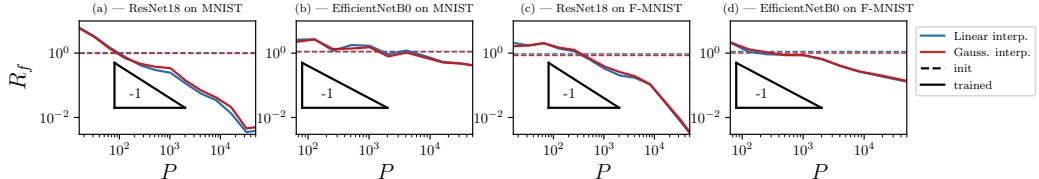

Figure 13: **Relative stability to diffeomorphisms $R_f(P)$ at $\delta = 1$.** Analogous to Figure 3-right but here we have MNIST (a-b) and FashionMNIST (c-d) in place of CIFAR10. Stability monotonically decreases with $P$. The triangles give a reference for the predicted slope in the stripe model – i.e. $R_f \sim P^{-1}$ – see Section 6. The slopes in case of ResNets are compatible with the prediction. For EfficientNets, the second panel of Fig.3 suggests that stability to diffeomorphisms is less important. Here, we also see that it builds up more slowly when increasing the training set size. Finally, blue and red colors indicate different interpolation methods used for generating image deformations, as discussed in Appendix C. Results are not affected by this choice.

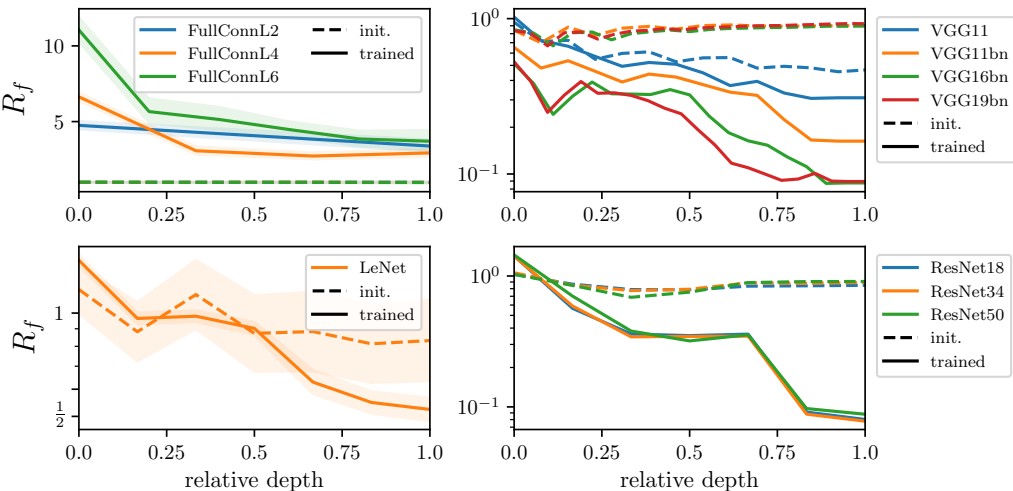

Figure 14: **Relative stability to diffeomorphisms as a function of depth.** $R_f$ as a function of the layers relative depth (i.e. $\frac{\text{current layer depth}}{\text{total depth}}$) where "0" identifies the output of the 1st layer and "1" the last. The relative stability is measured for the output of layers (or blocks of layers) inside the nets for simple architectures (1st column) and deep ones (2nd column) at initialization (dashed) and after training (full lines). All nets are trained on the full CIFAR10 dataset. $R_{f_0} \approx 1$ independently of depth at initialization while it decreases monotonically as a function of depth after training. *Statistics:* Each point is obtained by training 5 differently initialized networks; each network is then probed with 500 test samples in order to measure $R_f$. The results are obtained by log-averaging over single realizations.

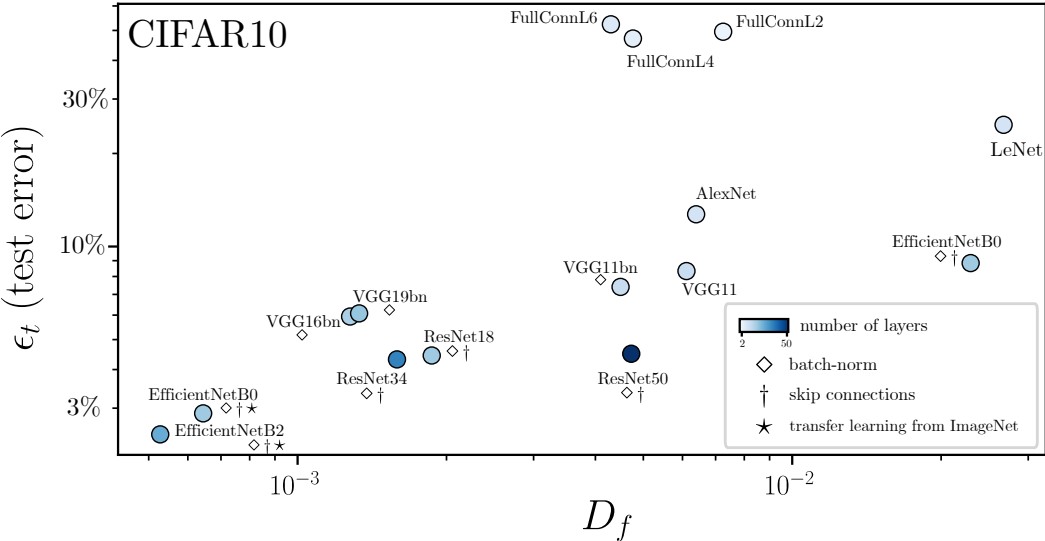

Figure 15: **Test error $\epsilon_t$ vs. stability to diffeomorphisms** $D_f$ for common architectures when trained on the full 10-classes CIFAR10 dataset ($P = 50k$) with SGD and the cross-entropy loss; the EfficientNets achieving the best performance are trained by transfer learning from ImageNet ($\star$) – more details on the training procedures can be found in Appendix E.1. The color scale indicates depth, and the symbols the presence of batch-norm ($\diamond$) and skip connections ($\dagger$). $D_f$ correlation with $\epsilon_t$ (corr. coeff.: 0.62) is much smaller than the one measured for $R_f$ – see Fig.3. *Statistics:* Each point is obtained by training 5 differently initialized networks; each network is then probed with 500 test samples in order to measure $D_f$. The results are obtained by log-averaging over single realizations.

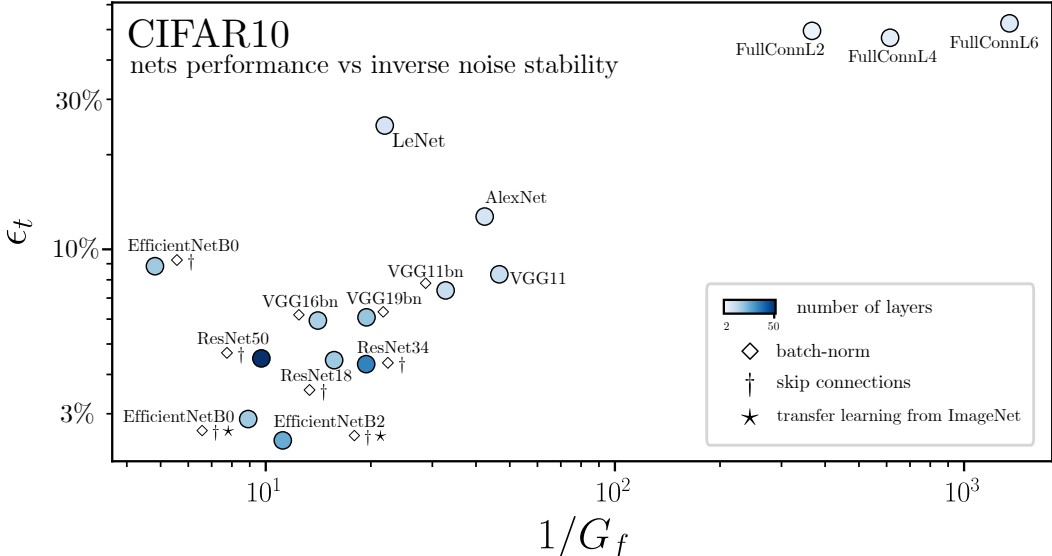

Figure 16: **Test error** $\epsilon_t$ **vs. inverse of stability to noise** $1/G_f$ for common architectures when trained on the full 10-classes CIFAR10 dataset ($P = 50k$) with SGD and the cross-entropy loss; the EfficientNets achieving the best performance are trained by transfer learning from ImageNet ($\star$) – more details on the training procedures can be found in Appendix E.1. The color scale indicates depth, and the symbols the presence of batch-norm ($\diamond$) and skip connections ($\dagger$). $G_f$ correlation with $\epsilon_t$ (corr. coeff.: 0.85) is less important than the one measured for $R_f$ – see Fig.3. *Statistics:* Each point is obtained by training 5 differently initialized networks; each network is then probed with 500 test samples in order to measure $G_f$. The results are obtained by log-averaging over single realizations.

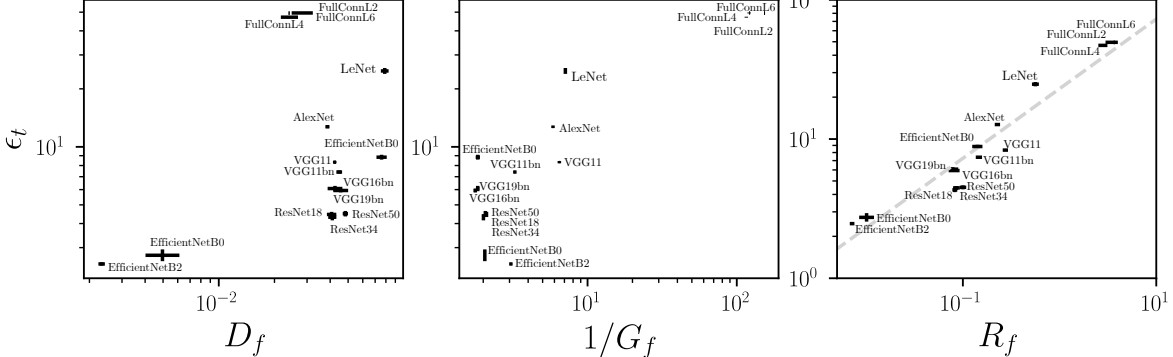

Figure 17: **Test error** $\epsilon_t$ **vs.** $D_f$**,** $1/G_f$ **and** $R_f$ **where** $\langle \cdot \rangle$ **is the mean.** Analogous to Figs15-19, we use here the mean instead of the median to compute averages over samples and transformations.

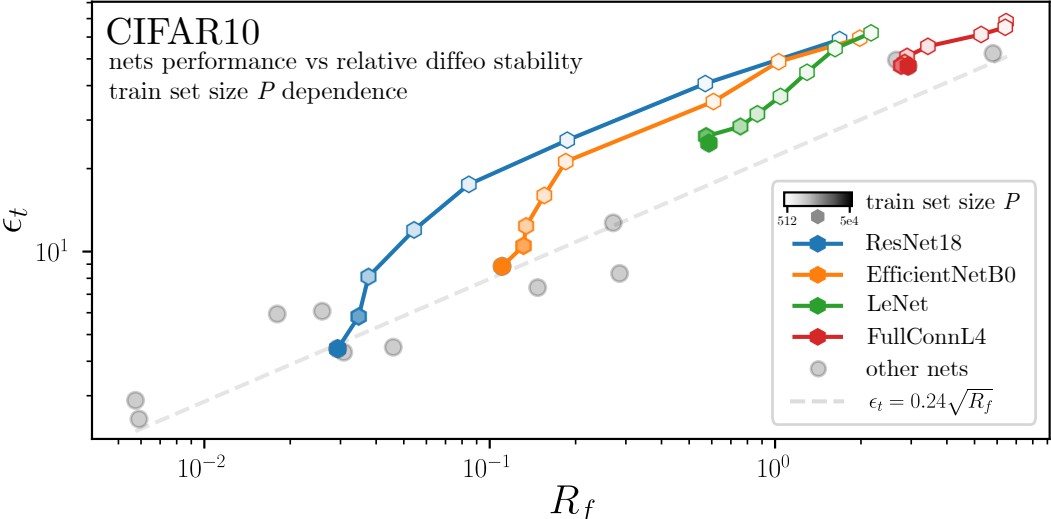

Figure 18: **Test error $\epsilon_t$ *vs.* relative stability to diffeomorphisms $R_f$ for different training set sizes $P$.** Same data as Fig.5, we report here curves corresponding to training on different set sizes for 4 architectures. The other architectures considered together with the power-law fit are left in background. For a small training set, CNNs behave similarly. *Statistics:* Each point is obtained by training 5 differently initialized networks; each network is then probed with 500 test samples in order to measure $R_f$. The results are obtained by log-averaging over single realizations.

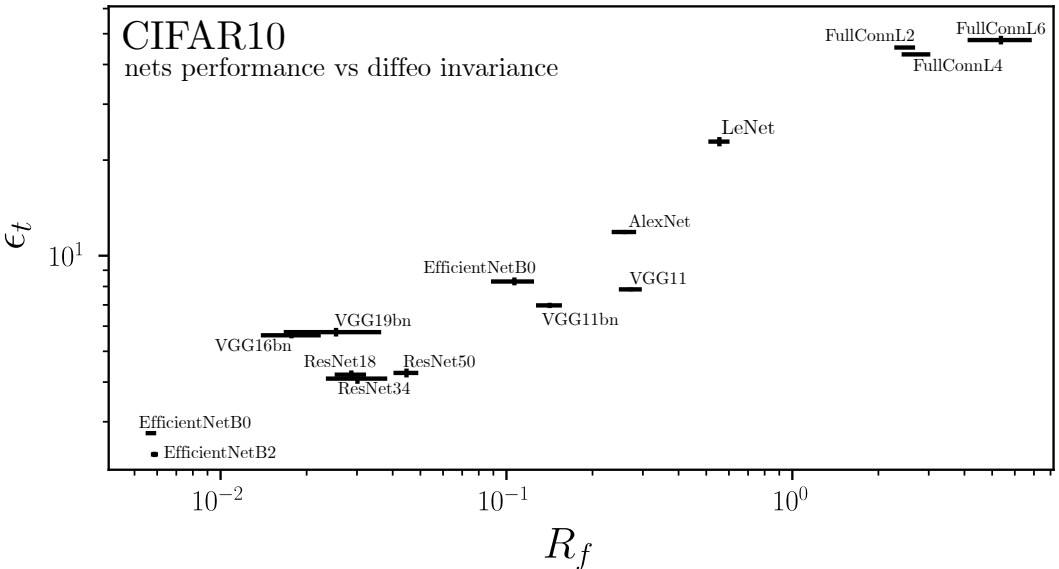

Figure 19: **Test error $\epsilon_t$ *vs.* relative stability to diffeomorphisms $R_f$ with error estimates.** Same data as Fig.5, we report error bars here. *Statistics:* Each point is obtained by training 5 differently initialized networks; each network is then probed with 500 test samples in order to measure $R_f$. The results are obtained by log-averaging over single realizations.

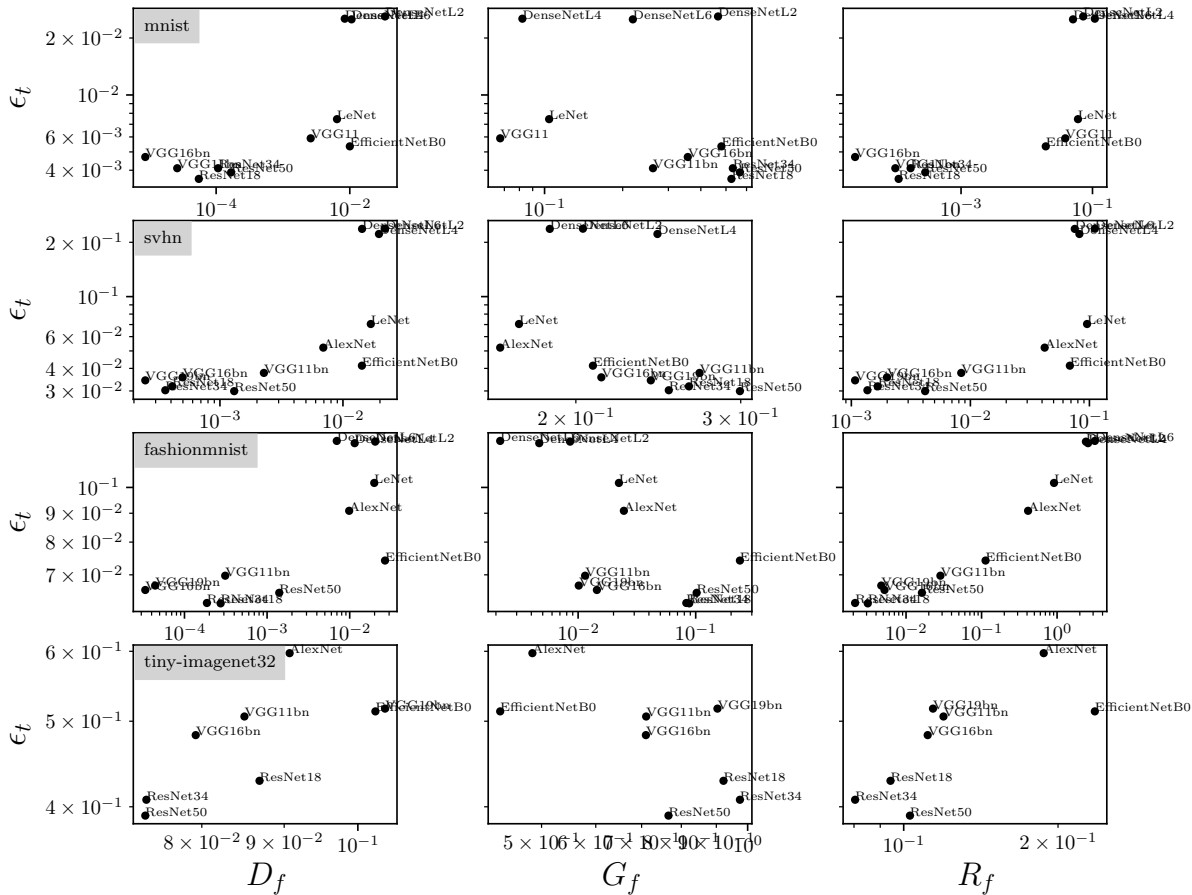

Figure 20: **Test error $\epsilon_t$ vs. $D_f$, $G_f$ and $R_f$ (on the columns) for different data sets (on the rows).** The corresponding correlation coefficients are shown in Table 2. Lines 1-2: MNIST and SVHN both contain images of digits and show a similar $\epsilon_t(R_f)$. Line 3: FashionMNIST results are comparable to the CIFAR10 ones shown in the main text. Line 4: Tiny ImageNet32 is a re-scaled (32x32 pixels) version of ImageNet with 200 classes and 100'000 training points. The task is harder than the other data sets and is such that we could not train simple networks (FC, LeNet) on it – i.e. the loss stays $\mathcal{O}(1)$ throughout training – so these are not reported here.

Table 2: **Test error vs. stability: correlation coefficients for different data sets.**

| data-set | $D_f$ | $G_f$ | $R_f$ |
|---|---|---|---|
| MNIST | 0.71 | -0.43 | 0.75 |
| SVHN | 0.87 | -0.28 | 0.81 |
| FashionMNIST | 0.72 | -0.68 | 0.94 |
| Tiny ImageNet | 0.69 | -0.66 | 0.74 |

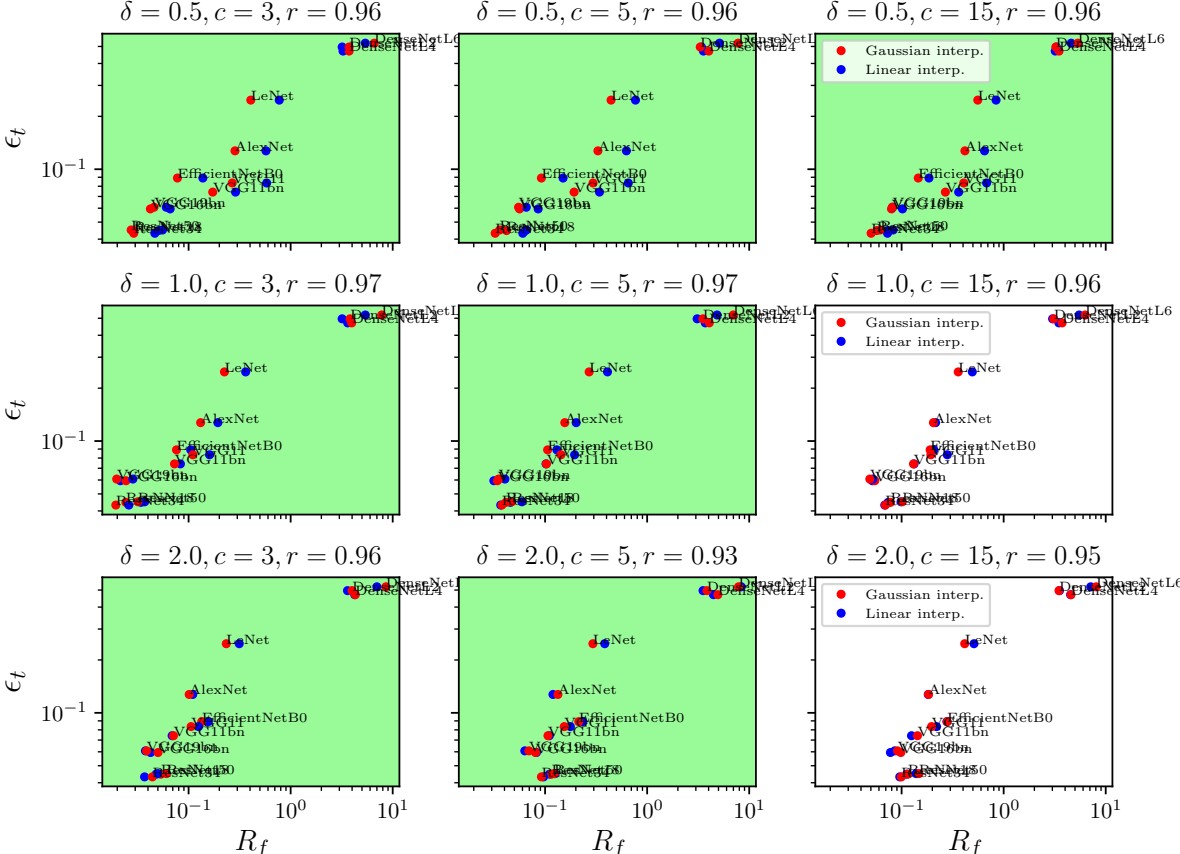

Figure 21: **Test error $\epsilon_t$ *vs.* $D_f$, $G_f$ and $R_f$ for CIFAR10 and varying $\delta$ and cut-off $c$.** Titles report the values of the varying parameters together with corr. coeffs. Parameters corresponding to allowed diffeo are indicated by the green background. Red and blue colors correspond to different interpolation methods. Overall, results are robust when varying these parameters.

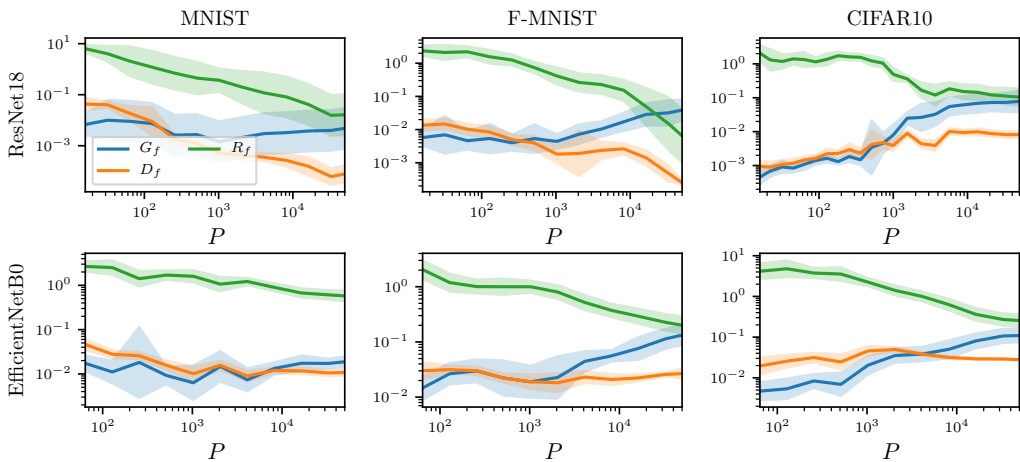

Figure 22: **Stability toward Gaussian noise ($G_f$) and diffeomorphisms ($D_f$) alone, and the relative stability $R_f$ with the relative errors.** Analogous to Fig.6 in which error estimates are omitted to favour clarity. Here we fix the cut-off to $c = 3$ and show error estimates instead. Columns correspond to different data-sets (MNIST, FashionMNIST and CIFAR10) and rows to architectures (ResNet18 and EfficientNetB0). Each panel reports $G_f$ (blue), $D_f$ (orange) and $R_f$ (green) as a function of $P$ and for different cut-off values $c$, as indicated in the legend. *Statistics*: Each point in the graphs is obtained by training 16 differently initialized networks on 16 different subsets of the data-sets; each network is then probed with 500 test samples in order to measure stability to diffeomorphisms and Gaussian noise. The resulting $R_f$ is obtained by log-averaging the results from single realizations. As we are plotting quantities in log scale, we report the relative error (shaded).