# OpenReview forum: "Relative stability toward diffeomorphisms indicates performance in deep nets"
_NeurIPS.cc/2021/Conference — NeurIPS 2021 Poster_

### Official Review · Reviewer_gEsS · 2021-07-11

**Rating:** 7
**Confidence:** 5

**Summary:**

This paper investigates the role of stability to the action of diffeomorphisms in the classification performances of deep neural networks. Surprisingly, it shows that stability measurements are good indicators to compare the performance of deep neural network classifiers even for complex image data bases. The paper introduces stability measurements of diffeomorphism actions over images and shows that these measurement correlate strongly with the classification errors obtained by different CNN architectures over MNIST and CIFAR. The paper also shows that the stability to diffeomorphisms decreases when the training sizes decreases and gives a simple model to explain this phenomenon.

**Limitations And Societal Impact:**

Yes

**Main Review:**

Stability to diffeomorphisms had been shown to have an important role  in order to establish appropriate similarity measures between images for classification, which are stable to deformations. However, has mostly been studied over non-convolutional neural networks based on wavelets or kernel representations. The remarkable element of this paper is to demonstrate that it is an important element present in all CNN for image classification. These results appear clearly in Figure 3 which demonstrates the increased stability to diffeomorphism actions of ResNEt et EfficientNet as the learning proceeds and in Figure 5 for CIFAR 10.  However, let us mention that the impressive result on Figure 5 is obtained for only one type of deformation for c=3 which is relatively lower frequency, one one data set (CIFAR 10).  However, larger displacement are a priori also valid and it would be interesting to verify wether the result holds when displacements are larger and with a larger c to see in what sense this property is robust.

The fact that learning stability to diffeomorphisms requires large data sets is also really interesting, specially when compared with stability with additive noise perturbations. The authors analysis indicating that precise network classifiers learns a sophisticated metrics which have large gradients and leading larger errors to additive white noise but small errors to action of diffeomorphisms is also very interesting. It is a first attempt to characterise the non-linear metrics constructed by such networks. In summary this is a very investing paper, demonstrating a new idea, that may lead to more fruitful research in this area.

**Time Spent Reviewing:**

4h

---

> ### Author Response · Authors · 2021-08-10
> **Authors response**
>
> We thank the reviewer for his/her comments.
>
> **Robustness of Fig.5:** Following the referee's suggestion, we confirm that the excellent correlation coefficient (corr) in Fig. 5 persists when the parameter c changes (corr=0,97; 0.98; 0.97 for c=3; 5; 15 respectively and displacement δ=1). It is also true when δ changes (corr=0.97; 0.93; 0.88 for δ=1; 2; 3 and c=3). Note that in this set-up, δ cannot be significantly larger than 3 as our ensemble then does not generate diffeomorphisms, see Section 2.2 (which may be responsible for the slight decrease of corr for δ=3).
>
> **Obtaining Fig. 5 for another data set:** to obtain such a figure, we had to train architectures, ranging from very modern to quite old, on the same data set.  Obtaining a similar figure for Imagenet (consisting of 10^6 images) will thus be extremely computationally intensive, as already trained primitive architectures  (such as fully-connected networks or LeNet) are not currently available. We thus analyzed FashionMNIST, where we can train the same architectures as in Fig.5. The results found are even more robust than CIFAR, with a correlation coefficient between test error and relative stability R included in [0.93, 0.97] as delta varies from 1 to 3, and c from 3 to 15. This observation strengthens our work and will be added to the paper- thanks!

---

> > ### Comment · Reviewer_gEsS · 2021-08-26
> > **Response to the authors**
> >
> > Thank you for your precisions. I maintain my ratings.

---

### Official Review · Reviewer_iL72 · 2021-07-16

**Rating:** 6
**Confidence:** 3

**Summary:**

The authors consider the relationship between the performance of modern deep
neural network architectures trained on standard benchmark natural image
datasets and their relationship to stability with respect to smooth
deformations of the input image. These issues have been studied previously in
the context of (natural) adversarial examples and in theoretically-constructed
(i.e. non-trainable) networks -- the motivation and contribution here is to
examine them through the lens of trained networks as well. To study this issue
empirically, the authors introduce a maximum-entropy distribution on
diffeomorphisms, which (I believe) amounts to constructing the deformation
vector field as a (gaussian) random combination of periodic Fourier basis
functions with frequency-dependent weights; they undertake an empirical
investigation of the properties of this distribution in order to find parameter
regimes where it produces realistic small-magnitude diffeomorphisms of the
image plane (rather than distorting it beyond recognition, or doing nothing).
They then use this tool to construct a computational "relative diffeomorphism
stability" metric, which is the average stability of a predictor with respect to
such diffeomorphisms divided by the predictor's average stability with respect
to scale-appropriate gaussian noise perturbations, called $R_f$. With $R_f$,
several findings are reported: across numerous modern architectures (ResNet18;
EfficientNet) and datasets (CIFAR10; ImageNet), randomly-initialized networks
have $R_f$ near unity and it decreases after training; $R_f$ decreases with the
size of the training set (measured by subsampling); and test set performance
correlates well with $R_f$ after training in the context of the CIFAR10
dataset. The authors present a toy theoretical model to give a theoretical
corroboration of the $R_f$ parameter's behavior for a shallow network, and
provide some ablation studies in the context of presence or not of data
augmentation in the training process and the type of interpolation used to
apply the diffeomorphisms.



**Limitations And Societal Impact:**

Yes.

**Main Review:**

## Summary wrt Decision

The paper studies an important problem in deep learning practice/theory that
seems to be somewhat underrepresented in the literature, namely the
relationship between a well-trained deep net's performance and its stability to
certain transformations of the input that specifically respect the structure of
natural image data. Progress on this issue may lead to new developments and
insights in (say) robust vision applications, in a way that other
theoretical/experimental investigations that do not specifically consider the
structure of the data in these applications may not. I have rated the paper as
I have due to certain confusions I have about the authors' interpretations of
their results and their significance; I would tend to increase the rating to
borderline accept or more if these are resolved (either by correcting me or
proposing clarifications) in the discussion phase.

I will provide more detailed comments below.

## Strengths
- The paper is very well-referenced, and code is provided for reproducibility.
  Most details are well-motivated theoretically, and the descriptive figures in
  the early sections give a good intuitive sense of the image plane
  diffeomorphisms that are being studied.
- The max-entropy distribution over diffeomorphisms is simple to compute and
  may well be useful more broadly as a tool for experimentation, as the authors
  point out.
- The theoretical component of the paper in section 6 is simple, but nontheless
  a good illustration of the central ideas of the paper. It could also be a
  good problem for future study (perhaps with deeper networks, more complicated
  data models, etc.)

## Issues

- I am unclear about the broad significance of the authors' findings, beyond
  the introduction of the maximum entropy distribution on diffeomorphisms as an
  experimental tool. I will frame this issue in terms of my understanding of
  the paper's motivation in the introduction -- understanding the relationship
  between *average case* deformation stability of modern deep networks and
  their generalization performance on benchmark datasets -- and on its stated
  implications for future research in the conclusion (section 7). Although the
  results in Figure 5 on correlation between $R_f$ and test error for CIFAR10
  are striking, and definitely more pronounced in terms of correlation
  coefficient than the corresponding plots for diffeomorphism stability and
  $\ell^2$ perturbation in Figs 15 and 16 in the appendices, the trends in
  Figures 15 and 16 are roughly the same -- the most performant modern
  architectures find themselves in the bottom-left of the plots, with a rough
  'upward' trend towards the top-right as one decreases the capacity of the
  network (modulo some outliers and inconsistencies). As a result I find the
  interpretation of these results in Sections 4 and 5 lacking -- section
  4 does not provide any interpretation of the results in Figure 5 beyond
  discussing some architectural/training factors, and the discussion in section
  5 is mostly restricted to considerations with respect to $P$ (which does not
  seem to be directly relevant to the results in Figure 5), and the important
  discussion in lines 228-238 seems hard to parse (I would appreciate some
  clarification here). It would be helpful if the authors interpreted these
  results in the context of discussions of (say) clean vs.\ robust accuracy in
  the literature, or more broadly other implications for practice. Without such
  interpretations, it is hard to see how the results in Figure 5 shed new light
  on these issues and may lead to further research progress on these important
  questions -- I note that the discussion of future directions in the
  conclusion is almost all concerning possible uses for the diffeomorphism
  ensemble, rather than implications of Figure 5.
- I am not sure about the broad significance of the discussion and experiments
  surrounding the training set size $P$. Although the stated conclusion that
  large training sets are necessary to get a small $R_f$ seems reasonable and
  useful, this is somewhat confounded by the fact that *diffeomorphism
  stability* is not as well correlated with large $P$ (section 5). It is
  therefore important to have a sense of the broad significance of a small
  $R_f$, as in the previous bullet. Also, the comparisons to the theoretical
  model of section 6 and the predicted $P^{-1}$ slope (Figure 3) might be
  spurious, as the fit is only 'good' in a compact region of the plot, whereas
  what really matters for such a claim to be accurate is the tail rate (and the
  datasets being experimented with obviously do not allow one to simulate out
  this far without relying on augmentations, which would not really serve the
  purpose).
- It would be nice to know if the authors' findings persist at larger dataset
  scales: although experiments on ImageNet are shown in Figure 3, the
  centerpiece Figure 5 (and corresponding plots in the appendices) are only for
  CIFAR10 and not ImageNet.
- It would be helpful throughout the paper if certain technical claims and
  implementation details were specified precisely, either in the body or in the
  appendices. On the technical side,  I found it difficult in
  certain cases to interpret which claims in section 2 are rigorous
  mathematical claims, which are heuristic (asymptotic; etc.) claims, and which
  are empirical claims: for example it would be nice if the appendix had a
  section summarizing all the calculations that are omitted in section 2, like
  "For large $n$, the norm then reads..." at line 108; the definition of
  $\delta$ and "It is straightforward to obtain..." at line 126; "that add
  pre-asymptotic terms to Eq.3" at line 128 (the fit seems not particularly
  good in fig 8a?). These details seem quite important to specify, because
  the introduction of the maximum entropy distribution on diffeomorphisms is
  claimed as a contribution here, and the implementation of this distribution
  seems from the discussion in section 2 and the appendix to have many finicky
  aspects that may be challenging to navigate if applying these methods to a
  new dataset (e.g. different image size or resolution or the like).
  On the implementation side, although it seems to be made more or less clear clear
  implicitly via Fig 2, the later discussions, and section D, it would be
  helpful in section 3 if it was stated exactly what $\eta$ is (I think it is
  additive independent gaussian noise on the image pixels, but there are
  certain vague/confusing statements around this term (line 138 "all random
  directions $\eta$"; figure 2 caption "For large $n$, it is equivalent to
  adding an i.i.d.\ Gaussian noise to all the pixel values of $x$" (one would
  like to know **what is being implemented** in the experiments, rather than
  how this fits into a theoretical formalism here)) that confound this
  interpretation.

## Questions, etc.

- In many specific vision applications, the types of diffeomorphism one is
  interested in obtaining invariance to with a deep net are far more structured
  than the generic near-identity transformations considered here: for example,
  in a registration application with a piecewise planar object undergoing a
  rigid body motion the transformations are homographies (e.g. [1]) and
  actually carry a low-dimensional structure. I understand the focus here is on
  generic transformation semi-invariance and how this relates to deep learning
  performance on standard benchmark datasets, but I am interested in whether
  the authors have considered measuring invariances to these types of
  structured transformations as well, and/or whether the maximum entropy
  formalism can be extended to this kind of setting. Such an extension could be
  of significant interest in practice, beyond generic 'small' diffeomorphism
  invariance.

## Minor issues (polishing, organization, etc.)

- Throughout section 3, following the definition of $R_f$, the authors
  repeatedly conflate having a small $R_f$ value with "stability to
  diffeomorphisms". It would seem to be more appropriate to add the qualifier
  "relative" in these cases, especially given the later claim (section 5) that
  stability to diffeomorphisms is "...not the good observable to characterize
  how deep nets learn invariance toward diffeomorphisms..."! In a similar vein,
  I found it somewhat jarring that $R_f$ was introduced at the start of section
  3 without a discussion of the natural complaint that this relative measure
  can be lowered significantly just by increasing the denominator -- given the
  prominence of $R_f$ in the results that will be interpreted in section 3
  immediately after this definition, I would think it appropriate to quickly
  mention at the end of the "Relative stability to diffeomorphisms" paragraph
  in section 3 that this issue will be discussed in detail later, in section 5
  (and perhaps foreshadow the conclusion, so the reader can trust that this is
  an interesting measure and process the figures in section 3).
- In lines 142-150, the choice $\delta = 1$ seems to me to be incompletely
  justified -- isn't the interpolation independence only significant for
  Resnet, and not EfficientNet? I might have read the figure wrong.
- The paragraph in lines 228-238 is somewhat difficult to parse; superficially
  it might be good to use a word other than "gradient" in this context since it
  is easily confused with the gradients one is using in training? (What seems
  to be meant here is large changes in the output of the predictor as a
  function of input; of course these can happen without a gradient existing.)

[1] http://arxiv.org/abs/1012.3216


**Time Spent Reviewing:**

5

---

> ### Author Response · Authors · 2021-08-10
> **Authors response**
>
> We thank the referee for his/her careful and useful review.
>
> **On the interpretation of our observations:**  a common belief is that stabilities to random noise (small G) and to diffeomorphisms (small D) are desirable properties of neural nets. Its underlying assumption is that the true data label mildly depends on such transformations when they are small. Our observations suggest an alternative view: (i) facts (Figs 6,16): better predictors are more sensitive to some small perturbations in input space (large G). (ii) Consequence: the notion that predictors are especially insensitive to diffeomorphisms is not captured by stability alone, but rather by the relative stability R=D/G. (iii) Interpretation for the correlation between R and performance: to perform well, the predictor must build large *gradients in input space* near the boundary decision (leading to a large G overall). Networks that are relatively insensitive to diffeomorphisms (small R) can (a) discover with less data that strong gradients must be there and (b) generalize them to larger regions of input space, improving performance and increasing G. The reviewer is right that this interpretation is not only in line with Fig.5,6 as stated in the text, but also consistent with Figs.15,16.
>
> This idea (iii) can be illustrated in the simple model of section 6, see Fig.7 Left panel. Imagine two data points of different labels falling close to the (say) left true boundary decision. These two points can be far from each other if their orthogonal coordinates differ. Yet if R=0 (now defined in Eq.6), then the output does not depend on the orthogonal coordinates, and it will need to build a strong gradient (in input space)  along the parallel coordinate to fit these two data. This strong gradient will exist throughout that entire boundary decision, improving performance but also increasing G. Instead if R=1, fitting these two data will not lead to a strong gradient (since they can be far from each other in input space). Beyond this intuition, in this model decreasing R can quantitatively be shown to increase performance, see Paccolat et al. (2021b).
>
> The comparison with adversarial robustness is not straightforward to us, since in our work we focus on typical directions in input space, instead of worst-case directions. Yet, our points (i,ii,iii) could be in line with the common observation that adversarial robustness is detrimental to performance, as the presence of strong gradients in input space may be detrimental to robustness if they are not perfectly aligned with the true boundary decisions (but still improve performance).
>
> Beyond our proposed interpretation, understanding more quantitatively the relationship between R and performance of Fig.5 for such data is a central question for the future, and a discussion will be added in conclusion (it was cut for reasons of length). For example, an interesting hypothesis is that the curve in Fig.5 delineates a “forbidden region” below it, where no algorithms can operate: a small R is necessary (clearly it is not sufficient) to obtain a certain level of performance. Introducing simplified models of data with controlled stability to diffeomorphisms would be useful to investigate this key question, beyond the very simple model of section 6.
>
> ***Action:*** we will clarify the paragraph with lines 228-238, by emphasizing its connection with Figs.15,16 as well as the link with the model of Section 6. We will add a paragraph in the conclusion to emphasize the challenge of understanding quantitatively Fig.5.
>
> **On Section 6:** We agree that the prediction of the very simple model may not hold quantitatively in deep nets, but as acknowledged by the referee and emphasized by us in the reply above, it is an interesting qualitative comparison to make.
>
> **On obtaining Fig.5 for other data sets:** see response to the last referee.
>
> **On Section 2:** mathematical statements are valid for asymptotically large c, and are obtained by replacing sums such as the one entering eq.2 by an integral. We will derive them in Appendix.
>
> **On η:** we will indicate that it is implemented by uniformly sampling the sphere (by rescaling an isotropic Gaussian noise).
>
> **Question:** A maximum entropy ensemble for finite dimensional transformations such as homographies can also be defined. Once a norm and a measure are chosen on these transformations, maximizing the entropy with a norm constraint to sample this ensemble can be done using a Metropolis Monte Carlo algorithm where the norm plays the role of an energy (in general, it may not have a simple analytical solution as in our case).
>
> **Minor issues:** We agree and will fix them. Concerning the choice δ=1, to answer the referee we have checked that the correlation coefficient of Fig.5 is 0.96 with the Gaussian interpolation (very similar to 0.98 with the bilinear interpolation). See also the response to Reviewer 4.

---

> > ### Comment · Reviewer_iL72 · 2021-08-30
> > **thanks**
> >
> > Dear authors,
> >
> > Thank you for your clarifications, especially around interpretation of your observations. The discussion and example you give makes sense to me and helps me appreciate the R measure. Your proposed additions to the discussion/conclusion seem to raise interesting questions to investigate in the future. I will increase my rating.

---

### Official Review · Reviewer_YhtQ · 2021-07-20

**Rating:** 6
**Confidence:** 3

**Summary:**

This paper introduces a new quantity called "relative stability" that strongly correlates with the classification performance of state of the art architectures on standard datasets. This quantity is a ratio between the variation of the features with respect to diffeomorphic transformation of the images and simple additive transformations.
To do so, the authors propose to use a parametrized set of diffemorphisms with bounded L2 norm of the derivative.
The dependence on the training set is highlighted, in particular a given amount of data is necessary to see the "relative stability" decrease with the size training set.
Last, a simple synthetic experiment is proposed that illustrates the expected behaviour.





**Limitations And Societal Impact:**

Since the work is essentially descriptive in nature there is not likely direct negative societal impact.

**Main Review:**

The main contribution of this work is twofold: (1) highlight the fact that stability to diffeomorphisms perturbations (or more precisely small deformation setting here) is not observed when increasing the dataset size. (2) introduce a quantity that is shown experimentally to decrease with the training size, at least for more advanced architectures.

The writing of the paper is clear and the main message (the relative stability indicator and its link with classification error) is new to my knowledge.

My opinion is that this contribution is definitively interesting however, it lacks a clear application, for instance in data augmentation, or a better theoretical understanding for a venue such as NeurIPS.


Other:
- There are many ways to generate diffeomorphisms (see references below and references therein). These have been used for data augmentation as is proposed in the conclusion of this paper.
- some references are missing:
   -- "Dreaming More Data: Class-dependent Distributions
over Diffeomorphisms for Learned Data Augmentation", by Hauberg et al.
   -- Anatomical Data Augmentation via Fluid-Based Image Registration, Shen et al.


Typos:
- page 5: "the an inverse"

**Time Spent Reviewing:**

3.5

---

> ### Author Response · Authors · 2021-08-10
> **Authors response**
>
> We thank the referee for finding our work definitely interesting and pointing out useful references. We respectfully disagree with his/her assessment that NeurIPS papers should always be theoretical or display a clear application. Testing existing theories and seeking new empirical laws is a cornerstone of natural sciences (there would be no Newtonian mechanics without Kepler’s laws). Example abounds showing that revealing empirical facts is central to the theoretical development of deep learning as well (including, in recent years, the observation of the double descent phenomenon, the fact that deep nets can learn randomly labeled data, that global minima of the loss landscape are connected, etc…). Here we introduce new tools to test existing ideas on the stability of diffeomorphisms, and to support a novel correlation between relative stability and performance. It has potential to affect our future understanding of this problem, as pointed out by the other referees.
>
> Our work has also potential for practical applications (including for data augmentation, briefly investigated in Appendix, and adversarial robustness), but mixing these topics with the fundamental questions we presently study would clearly be detrimental to the presentation of the latter.

---

> > ### Comment · Reviewer_YhtQ · 2021-08-23
> > **Response after rebuttal**
> >
> > I thank the authors for their answer. I agree with the authors that finding empirical laws can directly benefit the field. With hindsight, I also think that such papers have a potential to lead to fruitful research in comparison to incremental improvements of SOTA.
> > I am willing to change my rating accordingly to 6 (although a 5 would have been ok).
> >
> > As another comment on the main message of the paper: All the experiments have been done on standard datasets which matter in current benchmarks/applications. The hypothesis that the relative stability is data dependent has not been mentioned in the paper although it seems reasonable. Let me explain. As a thought experiment, if the dataset consists in diffeomorphic perturbations with random gaussian noise of a given shape and the classes are dependent on the size of the diffeomorphic perturbation, the learnt network (if it has good performance) should be sensitive to such perturbations, thereby inverting the ratio proposed by the authors. My opinion is that it is possible to achieve good performance with CNN based networks on such dataset.

---

> > > ### Author Response · Authors · 2021-08-24
> > > **Authors response**
> > >
> > > We thank the referee for his comment. We fully agree with him that our results hold for standard data sets of images, but will certainly not work for all data sets.
> > > For example, for data in which diffeomorphisms play no role (e.g. mixture of Gaussians of different labels with random centres), we don’t expect the relative stability R to
> > > evolve during training, nor to indicate performance. We will stress this point more in conclusion.

---

### Official Review · Reviewer_cGvy · 2021-07-21

**Rating:** 7
**Confidence:** 4

**Summary:**

This work contributes a framework for characterizing the invariance of image classifiers to diffeomorphisms. It is based on the empirical index $R_f$, which measures the stability to random diffeomorphisms relative to stability to additive noise. The authors find that $R_f$ is strongly correlated with classifier performance for a wide array of image classifiers. The results suggest that an image classifier's generalization performance is strongly related to its invariance to certain diffeomorphisms, as opposed to additive noise.

**Limitations And Societal Impact:**

Yes

**Main Review:**

The maximum entropy distribution of diffeomorphisms as well as the relative stability index $R_f$ were clearly explained, and the various figures made the ideas very easy to understand intuitively. The paper then thoroughly investigates the correlation between error and $R_f$ from various angles. In particular, the fact that $R_f$ is computed relative to noise makes the importance of diffeomorphisms more convincing. This paper has the potential to inspire further study of invariance to diffeomorphisms in deep nets, and the maximum entropy diffeomorphism distribution/$R_f$ measure are likely to be of independent interest for future work on data augmentation and adversarial robustness.

**Time Spent Reviewing:**

2

---

> ### Author Response · Authors · 2021-08-10
> **Authors response**
>
> We thank the reviewer for his/her comments.

---

### Decision · Program_Chairs · 2021-09-27

**Decision:**

Accept (Poster)

**Comment:**

All four knowledgeable reviewers recommend accepting this submission. I agree. This submission makes a valuable contribution by demonstrating the invariance of image classifiers to diffeomorphism.